# dCas13-mediated translational repression for accurate gene silencing in mammalian cells

Antonios Apostolopoulos [1,2], Naohiro Kawamoto [2], Siu Yu A. Chow[3], Hitomi Tsuiji[4], Yoshiho Ikeuchi [3,5,6], Yuichi Shichino [2] ✉ & Shintaro Iwasaki [1,2] ✉

Current gene silencing tools based on RNA interference (RNAi) or, more recently, clustered regularly interspaced short palindromic repeats (CRISPR)–Cas13 systems have critical drawbacks, such as off-target effects (RNAi) or collateral mRNA cleavage (CRISPR–Cas13). Thus, a more specific method of gene knockdown is needed. Here, we develop CRISPRδ, an approach for translational silencing, harnessing catalytically inactive Cas13 proteins (dCas13). Owing to its tight association with mRNA, dCas13 serves as a physical roadblock for scanning ribosomes during translation initiation and does not affect mRNA stability. Guide RNAs covering the start codon lead to the highest efficacy regardless of the translation initiation mechanism: cap-dependent, internal ribosome entry site (IRES)-dependent, or repeat-associated non-AUG (RAN) translation. Strikingly, genome-wide ribosome profiling reveals the ultrahigh gene silencing specificity of CRISPRδ. Moreover, the fusion of a translational repressor to dCas13 further improves the performance. Our method provides a framework for translational repression-based gene silencing in eukaryotes.

Since the discovery of RNA interference (RNAi)[1], posttranscriptional gene knockdown has been a common strategy for biological research and has shown therapeutic promise[2]. Although the small RNAs used for RNAi possess 21–22 nucleotide (nt)-long sequences, their target specificity relies mainly on the complementarity of 2–8-nt-long sequences at the 5′ end[3]. Thus, RNAi has the potential to silence off-target genes[4–6]. Moreover, due to the use of a double-stranded RNA duplex as a precursor of small interfering RNA (siRNA), the induction of an interferon response may be another issue[7–10]. Therefore, a method with higher accuracy and specificity has long been desired.

Recently, repurposing of the clustered regularly interspaced short palindromic repeats (CRISPR)-Cas system, a natural bacterial defense system against infecting viruses and plasmids, has allowed more specific RNA targeting. Cas13, a single type VI protein effector (class 2),

associates with a guide RNA (gRNA) containing a 20–30-nt-long spacer region with a direct repeat (DR) hairpin that anchors Cas13 proteins[11–16]. Hybridization of the spacer region to the target RNA activates Cas13 as an RNase[16–19]. The near-perfect complementarity between the spacer region and the target mRNA required for Cas13 activation is the basis of the superior specificity of this system compared to that of RNAi[17–22]. Cas13 gRNAs do not require the protospacer adjacent motif (PAM), which is necessary for Cas9 gRNAs, and have minimal restriction of the protospacer flanking sequence in human cells[17–19,23–25], allowing flexibility in the design. Although the primary substrate of Cas13 should be the target RNA that is directly bound, Cas13 also degrades neighboring RNA species indiscriminately due to the solvent exposure of two catalytic higher eukaryotes and prokaryotes nucleotide-binding (HEPN) domains[11–16]. This RNase activity toward bystander RNA species

[1]Department of Computational Biology and Medical Sciences, Graduate School of Frontier Sciences, The University of Tokyo, Kashiwa, Chiba 277-8561, Japan. [2]RNA Systems Biochemistry Laboratory, RIKEN Cluster for Pioneering Research, Wako, Saitama 351-0198, Japan. [3]Institute of Industrial Science, The University of Tokyo, Meguro-ku, Tokyo 153-8505, Japan. [4]Education and Research Division of Pharmacy, School of Pharmacy, Aichi Gakuin University, Nagoya, Aichi 464-8650, Japan. [5]Department of Chemistry and Biotechnology, School of Engineering, The University of Tokyo, Bunkyo-ku, Tokyo 113-0033, Japan. [6]Institute for AI and Beyond, The University of Tokyo, Bunkyo-ku, Tokyo 113-0033, Japan. ✉e-mail: yuichi.shichino@riken.jp; shintaro.iwasaki@riken.jp

(collateral activity) endows the immune system with Cas13[23,26]. Earlier reports suggested that this collateral activity is limited in eukaryotic cells[17–19]; indeed, subsequent application of this system for targeted knockdown aligned with these goals[20,22,27–46]. However, later studies have raised concerns that bystander RNA cleavage is unleashed in a wide variety of cells and leads to cytotoxicity[21,47–57], therefore posing an obstacle to the use of Cas13 for transcript-specific knockdown.

However, after the conversion of the catalytic residues to generate an inactive protein, Cas13 still serves as a platform to target RNA with high specificity. The tight (dissociation constant $K_d$ of 5–40 nM)[23,24,58–60] and specific association of catalytically inert Cas13 (dead Cas13 or dCas13) can be harnessed to impede RNA-binding protein association to control alternative splicing[19,61] and to track RNA mobility by GFP fusion[17,35,62–64] and by fluorophore-labeled gRNAs[65]. When fused to various effector proteins, dCas13 has also been used to induce A-to-I and C-to-U editing[18,52,66–72], m⁶A installation/removal[73–75], and proximity protein labeling on the defined mRNA[76,77]. This expanding toolkit paves the way for the spatiotemporal manipulation of target RNAs.

Here, we employed dCas13 to repress translation initiation in mammalian cells for accurate gene silencing. With the use of an RNase-inactive mutant, we reasoned that our system could circumvent the collateral activity of Cas13 in the transcriptome. Indeed, genome-wide ribosome profiling revealed the high specificity of our system for suppressing protein synthesis from the target mRNA. Tiling of the gRNAs along the target mRNA revealed the steric hindrance effect of dCas13 on the preinitiation complex during the scanning process and/or a stable association at the AUG start codon, ensuring the optimal design of the gRNAs (-23-nt length and start codon coverage). Our system applied not only to standard cap-dependent translation but also to internal ribosome entry site (IRES)-mediated and repeat-associated non-AUG (RAN) translation. Moreover, the fusion of the translational suppressor, eukaryotic translation initiation factor (eIF) 4E homologous protein (4EHP), further enhanced the silencing efficacy. Our method, termed CRISPRδ [delta (δ): DEpLetion of Translation by blockAde], provides a useful framework for a tool to implement precise repression of protein expression.

## Results

### dCas13 represses translation initiation of target transcripts

We hypothesized that the tight binding of Cas13 to an mRNA may abrogate the movement of ribosomes. Given the diversity of Cas13 family proteins[78,79], we aimed to survey their potential for translational repression (Fig. 1a). Considering their extensive applications in RNA biology[18,19,52,62–64,67–70,72,73,75–77], we generated catalytically inactive (dCas13) proteins corresponding to four selected Cas13 proteins: *Leptotrichia wadei* Cas13a (LwaCas13a and dLwaCas13a), *Prevotella* sp. *P5-125* Cas13b (PspCas13b and dPspCas13b), *Porphyromonas gulae* Cas13b (PguCas13b and dPguCas13b), and *Ruminococcus flavefaciens* XPD3002 Cas13d (RfxCas13d and dRfxCas13d). As subcellular localization has been reported to affect Cas13 performance[17,18,57,73,80], we expressed each of those Cas13 variants with a nuclear export signal (NES) or nuclear localization signal (NLS). NLS tagging of Cas13 variants may be advantageous for complex formation with transcribed gRNA before degradation, while NES tagging should be more reasonable for inhibiting translation, which occurs in the cytoplasm.

To evaluate the potential of dCas13 to repress translation in mammalian cells, we used a dual-reporter construct in human embryonic kidney (HEK) 293 cells. We transiently transfected a DNA plasmid encoding *Renilla* luciferase (Rluc) and firefly luciferase (Fluc), which are expressed as two different transcripts, and drove translation through a canonical cap-dependent mechanism (Supplementary Fig. 1a). This setup allows one luciferase to act as a direct target for dCas13, whereas the other luciferase serves as an internal control. For each dCas13 protein, we designed a gRNA with a 30-nt spacer targeting

the start codon of Rluc (Fig. 1b and Supplementary Fig. 1b). Transient coexpression of the Cas13/dCas13 protein, the respective gRNA, and the dual reporter allowed us to monitor the efficacy of gene expression repression.

The dCas13 effector reduced Rluc expression (Fig. 1c). The impacts of dCas13 expression were generally correlated with those of the wild-type (WT) variants but were weaker (Fig. 1c). Notably, RfxCas13d required the NLS to be more functional (Fig. 1c). Since, among the constructs tested, dPspCas13b fused to NES (dPspCas13b-NES) showed the most robust effect, we focused mainly on this variant for subsequent assays (Fig. 1c). We note that Cas13 variants exhibited differential expression levels (Supplementary Fig. 1c). Although NES-tagged RfxCas13d was the most inefficient for gene silencing (Fig. 1c), the expression of this protein was not the lowest (Supplementary Fig. 1c). In contrast, NES-tagged dPspCas13b was the most inefficiently expressed protein (Supplementary Fig. 1c) but exhibited the greatest repression of gene expression (Fig. 1c). Thus, protein expression explained the efficiency of the target expression reduction only partially.

While WT PspCas13b reduced Rluc mRNA levels, dPspCas13-induced Rluc repression was not associated with target RNA degradation (Fig. 1d); hence, this repression was considered to stem from the net reduction in translation. Importantly, translational repression was only induced when both the gRNA and dPspCas13 were expressed simultaneously (Supplementary Fig. 1d, e).

Given that dCas13 recruited to the start codon may physically obstruct the scanning 40S ribosome to impede translation initiation, we investigated the ability of dCas13 for inhibiting the elongating 80 S ribosome. Testing of gRNAs targeting various positions along the reporter transcript (Fig. 1e) revealed that dPspCas13b-NES targeting the open reading frame (ORF) (gRNA ORF #1, ORF#2, and stop codon) led to weaker translational repression (Fig. 1f). We also observed similar trends when dPspCas13b-NLS and dRfxCas13d-NLS were used for identical target sequences (Supplementary Fig. 1f, g). Notably, the defects in dCas13-mediated translational repression by the ORF-targeting gRNAs were not attributed to the inactivity of the gRNAs, since they exhibited robust knockdown capacity in the presence of WT Cas13 (Fig. 1f and Supplementary Fig. 1f, g). These data indicated that dCas13 may not pose a strong enough blockade to halt the highly processive 80S ribosome, which can displace RNA-binding proteins from mRNAs[81].

Considering the 40S scanning mechanism, we reasoned that, in addition to gRNAs covering the start codon, those targeting the 5′ untranslated region (UTR) may also induce translational repression. Thus, we designed gRNAs complementary to different positions in the 5′ UTR of the reporter (Fig. 1g). Although reporter assays revealed variation in the repressive capacity of dCas13 for the sequences targeted by the tested gRNAs (Fig. 1h), we observed translational repression for those gRNAs targeting the 5′ UTR. However, the gRNA targeting the start codon ensured the highest efficiency of translational repression. Experiments with dRfxCas13d-NLS led to conclusions similar to those reached for dPspCas13b-NES (Supplementary Fig. 1h). Therefore, we further optimized the gRNA context of the start codon in the dCas13-induced translational repression system.

### Construction of an optimal gRNA for translational repression

Given that gRNAs may have differential efficacy of target cleavage[18,24,80,82,83], we conducted a systematic survey to identify a gRNA design effective for translational inhibition. We tested the minimal length of the spacer region. Since the DR of the gRNA for PspCas13b is located at the 3′ end (Supplementary Fig. 1b), we retained the 3′ part of the spacer region and trimmed this region from the 5′ end (Fig. 2a). For example, gRNAs with short spacer regions containing 16 or 17 nt had diminished translational repression potency, and 19- to 25-

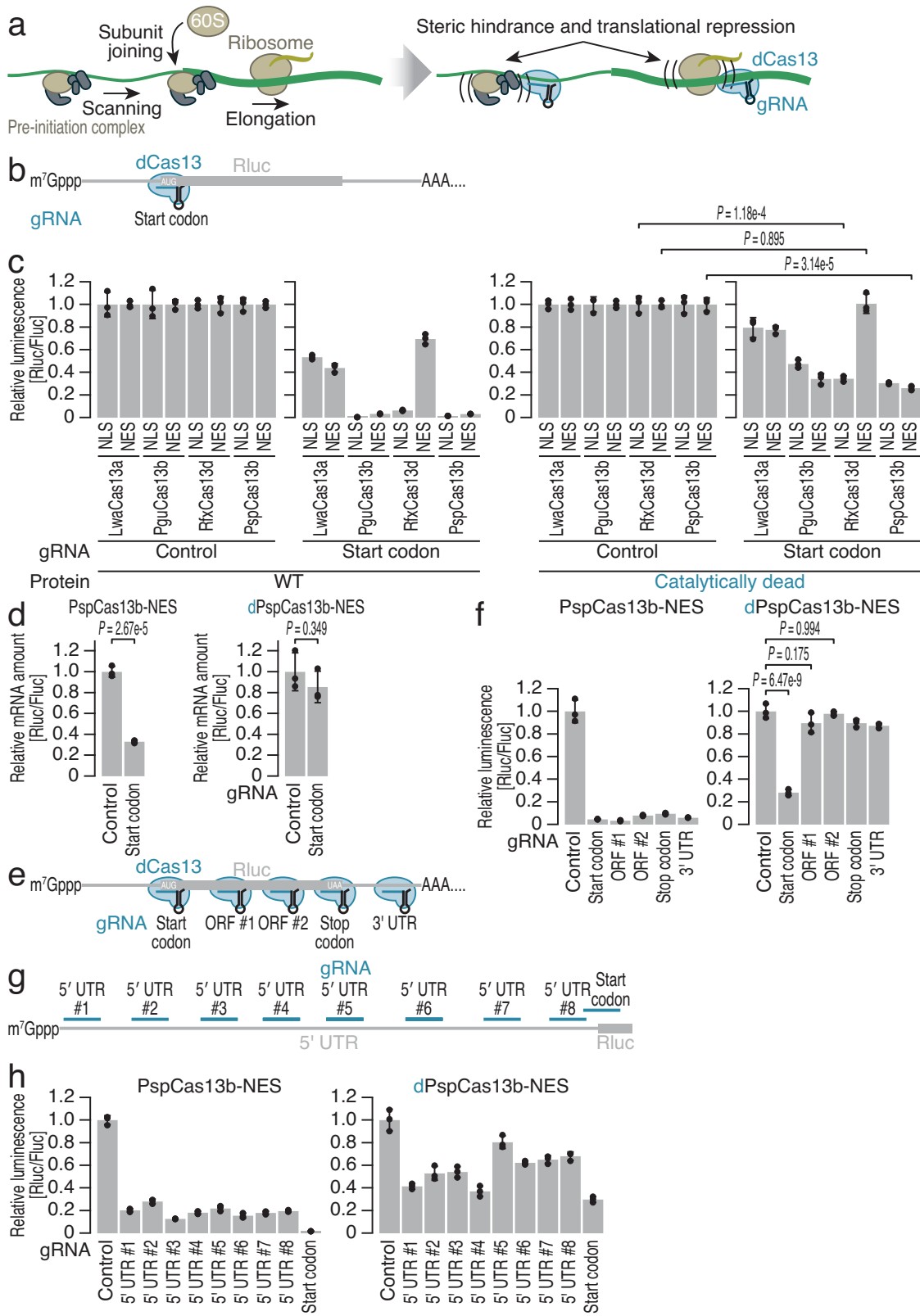

nt-long spacers provided the most efficient translational silencing (Fig. 2b and Supplementary Fig. 2a).

Since AUG masking served as a valid strategy, we sought to determine whether the position of the anti-start codon within the spacer region affects the translational repression capacity of dCas13. For this purpose, we tiled 30-nt-long gRNAs along the start codon of Rluc (Fig. 2c). Although we observed translational repression of Rluc with all the gRNAs tested, central placement of the anti-start codon (e.g., gRNA AUG9 and gRNA AUG17) in the spacer had a positive effect on translational repression (Fig. 2d and Supplementary Fig. 2b). A similar design principle was applied to other dCas13 variants (dPspCas13b-NLS and dRfxCas13d-NLS) (Supplementary Fig. 2c–f) and gRNAs targeting Fluc (Supplementary Fig. 2g, h).

**Fig. 1 | Repurposing catalytically inactive Cas13 proteins for translational repression. a** Schematic of dCas13-induced repression of mRNA translation. dCas13 binding to an mRNA can selectively repress its translation by sterically hindering the translational machinery. **b** Schematic of the luciferase reporter and gRNA design. gRNAs targeting the start codon of Rluc reporter mRNA were designed for each of the dCas13 and Cas13 variants. A gRNA containing a non-targeting spacer sequence was used as a control. Fluc expression was used as an internal control. The start codon-targeting gRNA for PspCas13b/dPspCas13b was the same as the gRNA AUG16 used in the experiments referenced in Fig. 2c, d. **c** Relative Rluc luminescence with respect to Fluc luminescence was calculated to quantify the repressive effects of WT Cas13 (left) and the dCas13 (right) variants using gRNAs targeting the start codon of Rluc, as shown in (**b**). **d** Relative Rluc

mRNA abundance with respect to Fluc mRNA abundance was quantified by RT–qPCR under the indicated conditions. **e, g** Schematic of the luciferase reporter assay and gRNA design. gRNAs targeting various positions along the Rluc reporter mRNA were designed for both PspCas13b-NES and dPspCas13b-NES. A gRNA containing a nontargeting spacer sequence was used as a control. Fluc expression was used as an internal control. **f, h** Relative Rluc luminescence with respect to Fluc luminescence was calculated to quantify the repressive effects of PspCas13b-NES (left) and dPspCas13b-NES (right) using the gRNAs shown in (**e, g**). In (**c, d, f**, and **h**) the mean (gray bar), s.d. (black line), and individual replicates ($n = 3$, black points) are shown. In (**c, d**, and **f**), the $p$ values were calculated by Student's $t$ test (two-tailed) (**c, d**) and by the Tukey–Kramer test (two-tailed) (**f**).

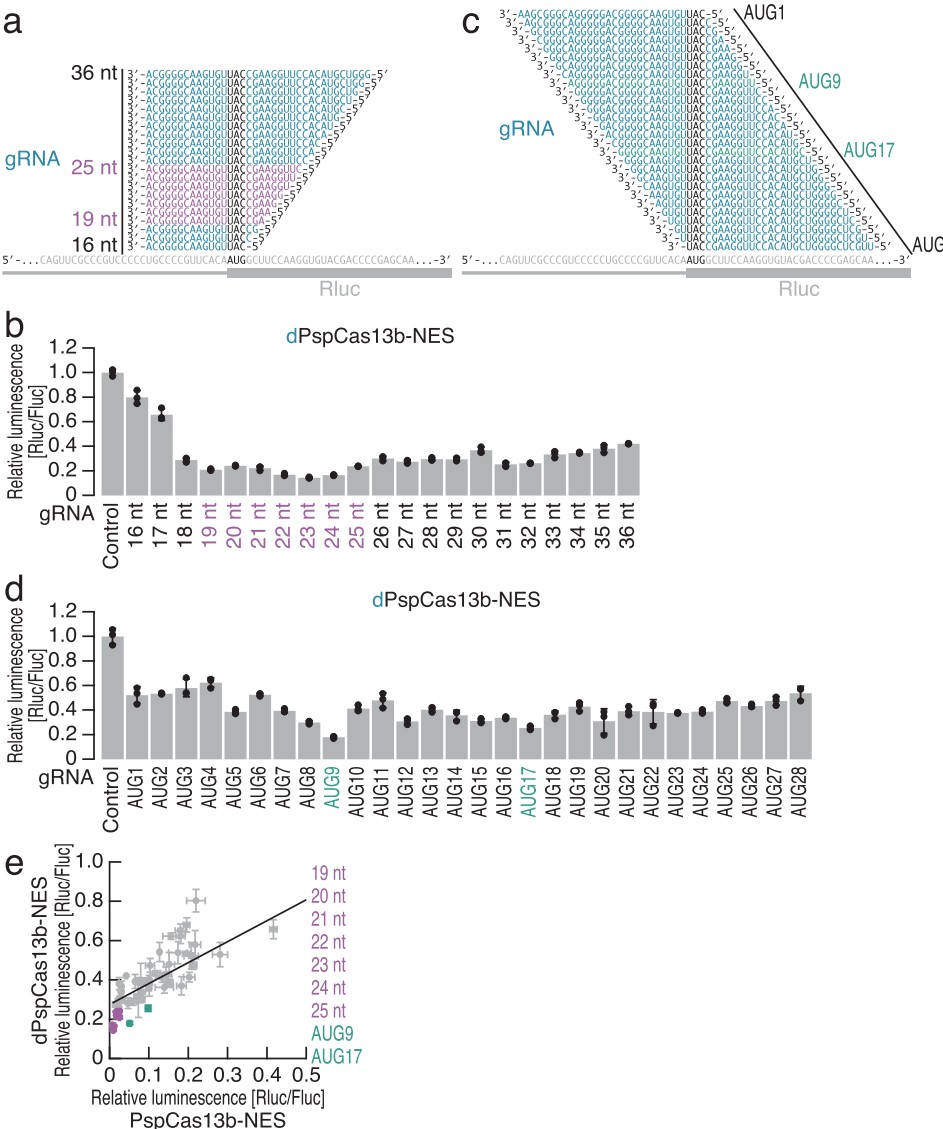

**Fig. 2 | Optimal gRNA design for CRISPRδ.** Schematic of the gRNA design for targeting the start codon of Rluc reporter mRNA. The gRNAs with the highest efficacies are highlighted in purple (**a**) and green (**c**). **b, d** Relative Rluc luminescence with respect to Fluc luminescence was calculated to quantify the repressive effect of dPspCas13b-NES using the gRNAs shown in (**a, c**). The mean (gray bar), s.d. (black line), and individual replicates ($n = 3$, black points) are shown. The gRNAs

with the highest efficacies are highlighted in purple (**b**) and green (**d**). **e** Comparison of the repressive effects of PspCas13b-NES and dPspCas13b-NES across the gRNAs used in this study. The mean (gray point, $n = 3$), s.d. (gray line), and regression line (black line) are shown. The gRNAs with the highest efficacies are highlighted in purple and green. The data from Figs. 1f, h, 2b, d were replotted.

Generally, the repression potential of dPspCas13b-NES across the tested gRNAs scaled with that of the WT counterpart (Fig. 2e), indicating that the efficacy of translational repression relies on the accessibility of the complex to the mRNA, as is the case for WT

Cas13[80,82,83]. However, the gRNAs with short spacers (19-25 nt) and those masking the AUG start codon in the center of the spacer (positions 9 and 17) exhibited an advantage in translational repression (Fig. 2e).

This translational repression strategy by dPspCas13b was effective not only in the adherent cells (HEK293 cells) used in the experiments above but also in suspension cells (HeLa S3 suspension, Supplementary Fig. 2i) and primary culture cells [mouse embryonic fibroblasts (MEFs), Supplementary Fig. 2j].

Considering these observations, we termed our method of dCas13-mediated translational silencing CRISPRδ.

## The high specificity of CRISPRδ

Given that WT Cas13 shows collateral RNase activity in mammalian cells[21,47–57], we investigated the specificity of CRISPRδ. For this purpose, we performed ribosome profiling, a technique for deep sequencing of ribosome-protected RNA fragments generated by RNase treatment[84,85], and RNA-Seq of the same samples. We applied this approach in a strain of HEK293 cells expressing enhanced GFP (EGFP) from a stable genomic integrant, with transient expression of Cas13 proteins and gRNAs. Here, we designed gRNA to target EGFP (Fig. 3a) in accordance with the configuration of the effective gRNA in the luciferase-based assay (Fig. 2). Since the stochastic expression of Cas13 proteins in cells (Supplementary Fig. 3a, b) hampers quantitative analysis, we sorted cells by flow cytometry to ensure the expression of mCherry-tagged Cas13 in the cells. Indeed, we observed that the cells with higher expression of mCherry-tagged Cas13 (WT or catalytically dead) had more potent repression of EGFP (Supplementary Fig. 3a, c).

Combined fluorescence-activated cell sorting (FACS) and ribosome profiling/RNA-Seq were performed to evaluate the effects of the Cas13 variants on gene expression in a genome-wide manner (Fig. 3b). The hallmarks of the read features, such as the peak footprint lengths of 29-30 nt and 21-22 nt (Supplementary Fig. 3d)[86,87] and the 3-nt periodicity along the ORF (Supplementary Fig. 3e), validated the quality of the ribosome profiling in the cells collected by FACS. As reported in earlier studies[21,47–57], the collateral activity of WT PspCas13b-NES induced changes in the expression of nontarget mRNAs (987 upregulated transcripts and 1498 downregulated transcripts), in addition to the intended reduction in EGFP expression (Fig. 3c left). In contrast, dPspCas13b-NES did not lead to any transcriptomic changes (including EGFP mRNA) (Fig. 3d left). However, in terms of translation efficiency—that is, the over- or underrepresentation of ribosome footprints with respect to the changes in RNA abundance—dPspCas13b-NES suppressed EGFP expression (Fig. 3d right). Importantly, this suppression was highly specific for this gRNA-targeted transcript; we found no significant alterations in mRNA expression across the transcriptome (Fig. 3d right).

Notably, even WT PspCas13b-NES may induce translational alterations in the target EGFP transcript and other transcripts (Fig. 3c right). Translational repression of EGFP probably occurs because the recruitment of the WT variant also results in steric hindrance of scanning ribosomes before transcript degradation. In addition, ribotoxic stress[88,89] due to partial cleavage of ribosomal RNA (rRNA) by WT Cas13[55] may elicit changes in the translation of nontarget mRNAs. Again, we did not find that dPspCas13b-NES exerted such a nonspecific effect on translation (Fig. 3d right).

As a result of WT PspCas13b- and dPspCas13b-mediated gene silencing, we also observed a reduction in EGFP protein by Western blotting as well (Fig. 3e and Supplementary Fig. 3f).

Taken together, these data led us to conclude that the CRISPRδ method can be used for translational repression with ultrahigh specificity.

## CRISPRδ represses cap-independent translation

Although most mRNAs utilize the cap-dependent process to initiate translation in eukaryotic cells[90], a subset of mRNAs exploit cap-independent mechanisms such as internal ribosome entry site (IRES)-mediated translation[91,92]. To explore the ability of CRISPRδ to repress IRES-mediated protein synthesis, we used an Rluc reporter fused to the hepatitis C virus (HCV) IRES[93,94] (Supplementary Fig. 4a), which directly recruits the 40S ribosome to the AUG codon and bypasses the cap-binding protein eIF4E, scaffold protein eIF4G, and RNA-binding protein eIF4A. Indeed, treatment with hippuristanol, an eIF4A inhibitor[95,96], confirmed that our reporter was translated independently of the protein (Supplementary Fig. 4b).

dPspCas13b-NES repressed translation driven by the HCV IRES (Figs. 4a, b). This effect was specific for the gRNA targeting the AUG codon of the HCV IRES. The downstream AUG codon, which was initially used as the start codon of Rluc (Fig. 4a), could not serve as a valid target (Fig. 4b), corroborating the idea that dCas13 is not potent for blocking translation elongation.

We also used another IRES from the intergenic region (IGR) of the cricket paralysis virus (CrPV) (Fig. 4c and Supplementary Fig. 4c)[97,98], which does not require any translational initiation factors. In contrast to the results of DNA transfection for reporter expression in the experiments above, we introduced in vitro synthesized mRNA with an A cap (Supplementary Fig. 4c), which suppressed cap-dependent translation, into cells. The nanoluciferase (Nluc)-based reporter was ensured to be driven by eIF4A-independent mechanisms (Supplementary Fig. 4d). dPspCas13b-NES inhibited CrPV-mediated translation (Fig. 4d), even though the uncanonical GCU start codon was used in this context.

Although the secondary structure of IRES and the interacting factors may ultimately affect the efficiency, CRISPRδ serves as a useful tool to suppress cap-independent translation.

## Translational silencing of pathogenic RAN translation from ALS- and FTD-linked *C9orf72*

Given the potency of our method for translation repression, we next tested its ability to suppress pathological translation. A hexanucleotide (GGGGCC) repeat expansion in an intronic region of chromosome 9 open reading frame 72 (*C9orf72*) is the most frequent cause of frontotemporal dementia (FTD) and amyotrophic lateral sclerosis (ALS)[99,100]. One potential mechanism by which this mutation can cause disease is that the expanded GGGGCC repeat tract becomes a substrate for an unconventional form of translation called repeat-associated non-AUG (RAN) translation[101]. RAN translation drives protein synthesis from multiple reading frames in GGGGCC repeats, producing dipeptide repeat proteins (DPRs) that contain repeats such as Gly-Arg (GR), Gly-Ala (GA), and Gly-Pro (GP). RAN translation from the antisense transcript possessing GGCCCC repeats also generates DPRs with Pro-Arg (PR), Pro-Ala (PA), and Gly-Pro (GP) repeats. These DPRs are prone to aggregation and accumulate in the brain and spinal cord of patients harboring *C9orf72* mutations[102–105]. Thus, silencing RAN translation could be a powerful therapeutic approach for FTD/ALS as well as other neurodegenerative diseases in which RAN translation plays a role (*i.e.*, Huntington's disease, ataxia, and others)[106].

We applied CRISPRδ to a RAN translation reporter system harboring 66 *C9orf72* GGGGCC repeats[107]. We fused the GGGGCC repeats to the short HiBiT tag in three different frames to monitor poly(GR), poly(GA), and poly(GP) translation (Fig. 5a). The introduction of this construct did not influence cell viability due to the low expression of these genes under our conditions (Supplementary Fig. 5a). We designed gRNAs specific for the reported CUG subcognate start codon (gRNA start codon) for poly(GA)[108–114] and the upstream 5′ UTR (gRNA 5′ UTR #1 and 5′ UTR #2) (Fig. 5a). Although the translation start sites for poly(GR) and poly(GP) have not been well defined[108–114], these gRNAs repressed translation from all three DPR frames (Fig. 5b−d and Supplementary Fig. 5b).

Because G-quadruplex structures formed by GGGGCC repeats have been demonstrated to impact RAN translation[110,115], we investigated the effect of direct dCas13 recruitment to GGGGCC repeats. We tested other sets of gRNAs with perfect complementarity to single loci at the edges of GGGGCC repeats (gRNA Unique #1 to #7) and that

 

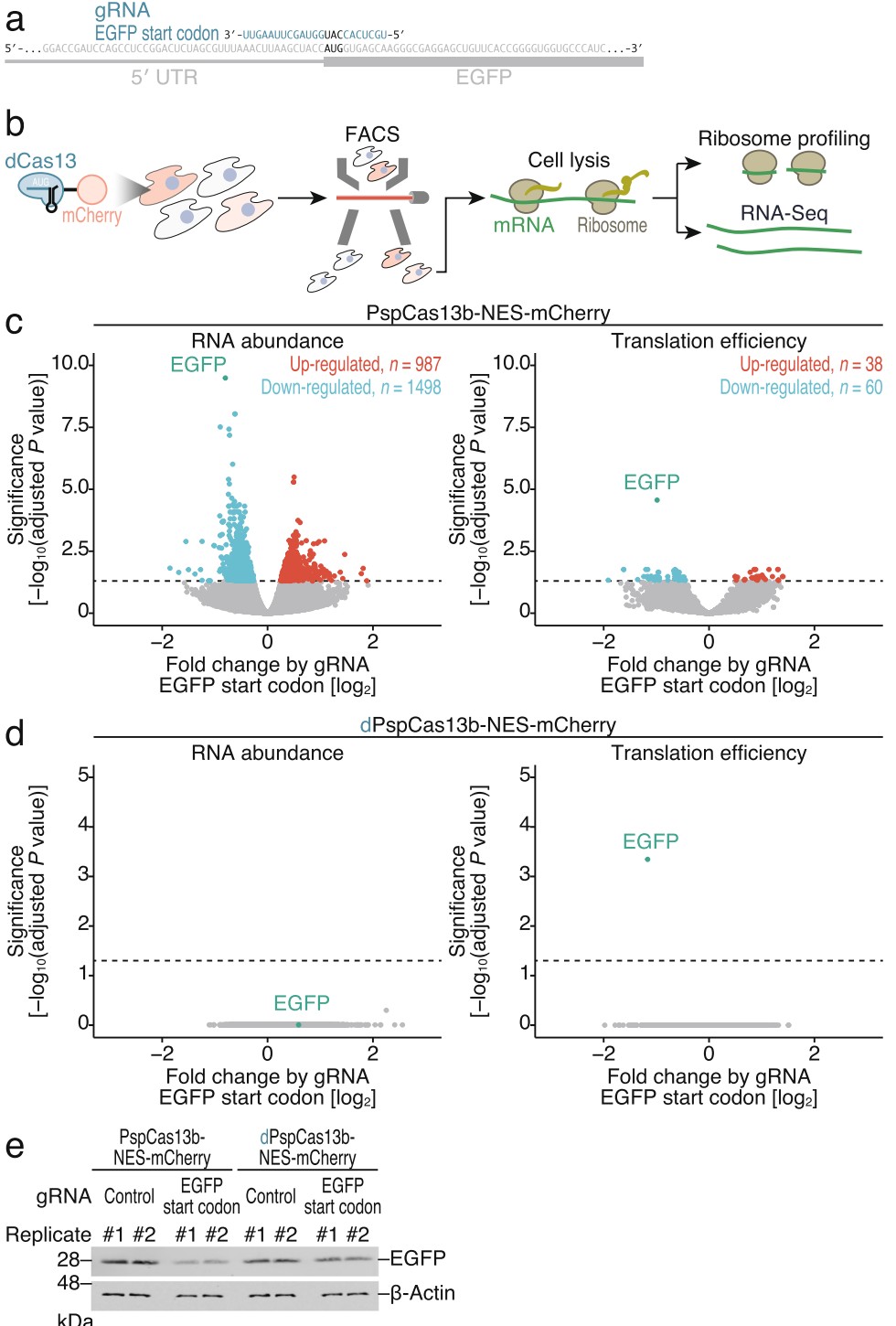

**Fig. 3 | Ribosome profiling and RNA-Seq reveal the high specificity of CRISPRδ for gene silencing. a** Schematic of the gRNA design for targeting the start codon of the EGFP reporter mRNA. **b** Schematic of the experimental procedures for ribosome profiling and RNA-Seq. PspCas13b variant expression was ensured by cell sorting. **c** Volcano plots of significance and fold changes in RNA abundance (left) and translation efficiency (right) according to the expression of the gRNA targeting the EGFP start codon and WT PspCas13b-NES. mRNAs with significant alterations (adjusted *p* value < 0.05) are highlighted. The *p* values were calculated by the likelihood ratio test (two-tailed) and adjusted by Benjamini–Hochberg method. **d** Same as (**c**) but for dPspCas13b-NES. **e** Western blotting of the indicated proteins. β-Actin was used as a loading control. Two replicates were tested.

targeted multiple sites within GGGGCC repeats (gRNA Multiple #1 to #6) (Fig. 5a). Some of these gRNAs reduced the expression of DPRs (Fig. 5b–d and Supplementary Fig. 5c–d). This effect might be caused by modulation of the G-quadruplex by dCas13, similar to what occurs for other RNA-binding proteins[115]. Another possibility is that dCas13 may mask translation initiation sites hidden in GGGGCC repeats, such as those for poly(GR) and poly(GP). Alternatively, dCas13 may be able to impede slow ribosome traversal on DPR ORFs[114,116], even though dCas13 cannot serve as a strong physical obstacle to translation elongation (Figs. 1f, 4b).

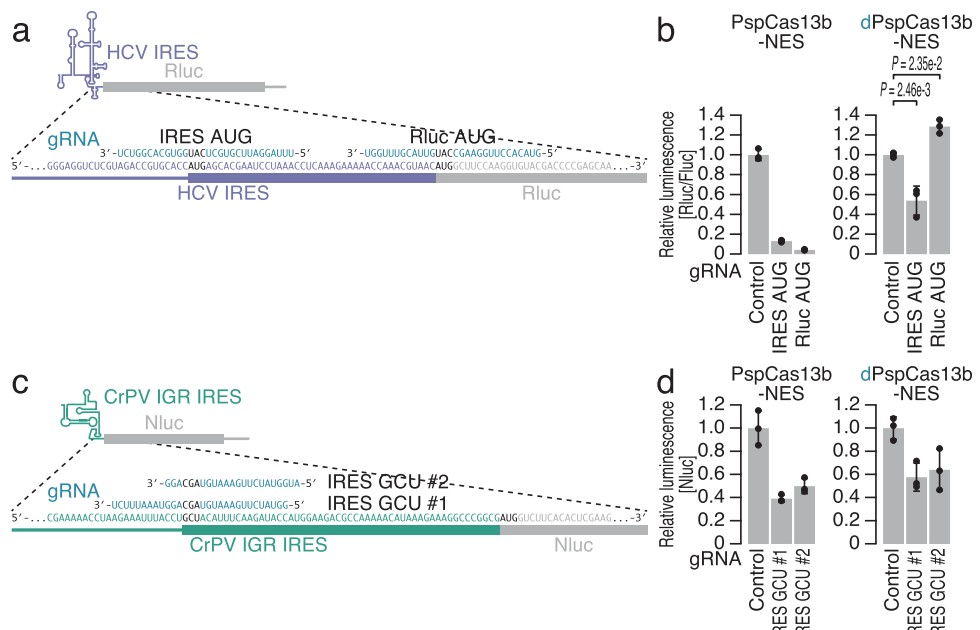

**Fig. 4 | CRISPRδ represses IRES-driven translation. a** Schematic of the gRNA design for targeting the start codon of Rluc reporter mRNA containing the HCV IRES. **b** Relative Rluc luminescence with respect to Fluc luminescence was calculated to quantify the repressive effects activity of dPspCas13b-NES using the gRNAs shown in (**a**). **c** Schematic of the gRNA design for targeting the start codon of Nluc reporter mRNA containing the CrPV IGR IRES. **d** Relative Nluc luminescence was calculated to quantify the repressive effects of dPspCas13b-NES using the gRNAs shown in (**c**). In (**b**) and (**d**), the mean (gray bar), s.d. (black line), and individual replicates (*n* = 3, black points) are shown. In (**b**), the *p* values were calculated by the Tukey–Kramer test (two-tailed).

We note that since some gRNAs may increase DPR synthesis (Fig. 5b–d and Supplementary Fig. 5c, d), careful design and selection of gRNAs should be important (e.g., gRNA Unique #3, #5, and #7 should be avoided). This opposite effect may be the outcome of the complicated effects of DPRs on global translation[108,117] and mRNA stability[118]. Related to this, we observed that, in addition to WT PspCas13b, dPspCas13b also induced a reduction of target mRNAs (Supplementary Fig. 5e). Within the set of conditions examined (Figs. 1d, 3d and Supplementary Fig. 6c, f), this phenomenon was the sole exception. Since DPRs synthesized from *C9orf72* GGGGCC repeats suppress RNA exosome activity and stabilize transcripts[118], dCas13-mediated translational repression may lead to activation of the mRNA decay pathway and destabilization of reporter mRNAs.

Together, these data provide evidence that CRISPRδ can be harnessed to suppress pathogenic protein synthesis independent of template RNA degradation and raise the possibility of employing CRISPRδ as a therapeutic strategy for nucleotide-repeat diseases such as *C9orf72* FTD/ALS.

### Enhanced CRISPRδ system

Despite the high specificity of CRISPRδ with dCas13, the efficacy of this approach was modest compared to that of WT Cas13. Whereas WT Cas13 can catalytically degrade target mRNAs in multiple rounds of reactions, dCas13 is required to associate with the target mRNA for a long enough duration to suppress translation and thus needs to be stoichiometrically abundant. To enhance the effect of dCas13 on translational repression, we tested multiple gRNAs targeting a single mRNA; two different gRNAs were used to increase the duration of dCas13-mRNA association (Supplementary Fig. 6a). This strategy enabled further downregulation of target mRNA translation by gRNAs (Supplementary Fig. 6a), which originally showed modest translational repression activity.

We also tested the fusion of a translational repressor to dCas13. This option has been successfully used to boost dCas9-mediated transcriptional repression via CRISPR interference (CRISPRi)[119–122].

Considering the steric hindrance effect of dCas13 on scanning 40S ribosomes, another mechanism of translational repression by the fused protein may further benefit translational repression (Fig. 6a). Here, we tested several translational suppressor proteins, including programmed cell death 4 (PDCD4)[123–126], eIF4E-transporter protein (4E-T)[127–129], eIF6[130,131], 14-3-3σ[132], poly(A)-binding protein-interacting protein 2 (PAIP2)[133–138], Pelota (PELO)[139–143], and 4EHP[144–150], on dPspCas13b-NES, which targets the start codon of the reporter, and found that 4EHP had the most significant effects (Supplementary Fig. 6b). The recruitment of 4EHP-fused dPspCas13b-NES not only to the start codon but also to the 5′ UTR outperformed that of the nonfused variant (Fig. 6b). Moreover, this fusion system allowed the inhibition of protein synthesis with a gRNA targeting the 3′ UTR, which originally could not impact translation with non-fused dCas13 (Figs. 1f, 6b). Similar artificial tethering of 4EHP to the 3′ UTR (*e.g.*, BoxB hairpin RNA/λN tag and MS2 stem–loop/MS2 binding protein) was reported to induce translational repression[147,150]. dCas13 serves as a scaffold for tethering translational repressors to defined regions of specific mRNAs. 4EHP-enhanced translational repression by dCas13 was not associated with mRNA instability (Supplementary Fig. 6c). We noted that the enhanced translational repression with 4EHP fusion was attributed to neither the global translation change, as measured by newly synthesized proteins labeled with *O*-propargyl-puromycin (OP-puro) (Supplementary Fig. 6d), nor to cell viability alteration (Supplementary Fig. 6e).

### CRISPRδ silences endogenous mRNA translation

To test the potency of the enhanced CRISPRδ system, we applied it to endogenous cellular mRNAs. Here, we designed three gRNAs targeting the CD46 start codon (Fig. 6c) for the PspCas13b variants and measured the abundance of the CD46 protein on the cell surface via flow cytometry. Again, we focused on the subpopulation of HEK293 cells in which mCherry-tagged PspCas13b expression was ensured. In addition to WT PspCas13b-NES (Fig. 6d), dPspCas13b-NES suppressed CD46 protein expression (Fig. 6e). The efficacy was further augmented by 4EHP fusion (Fig. 6f, g, gRNA CD46 #1).

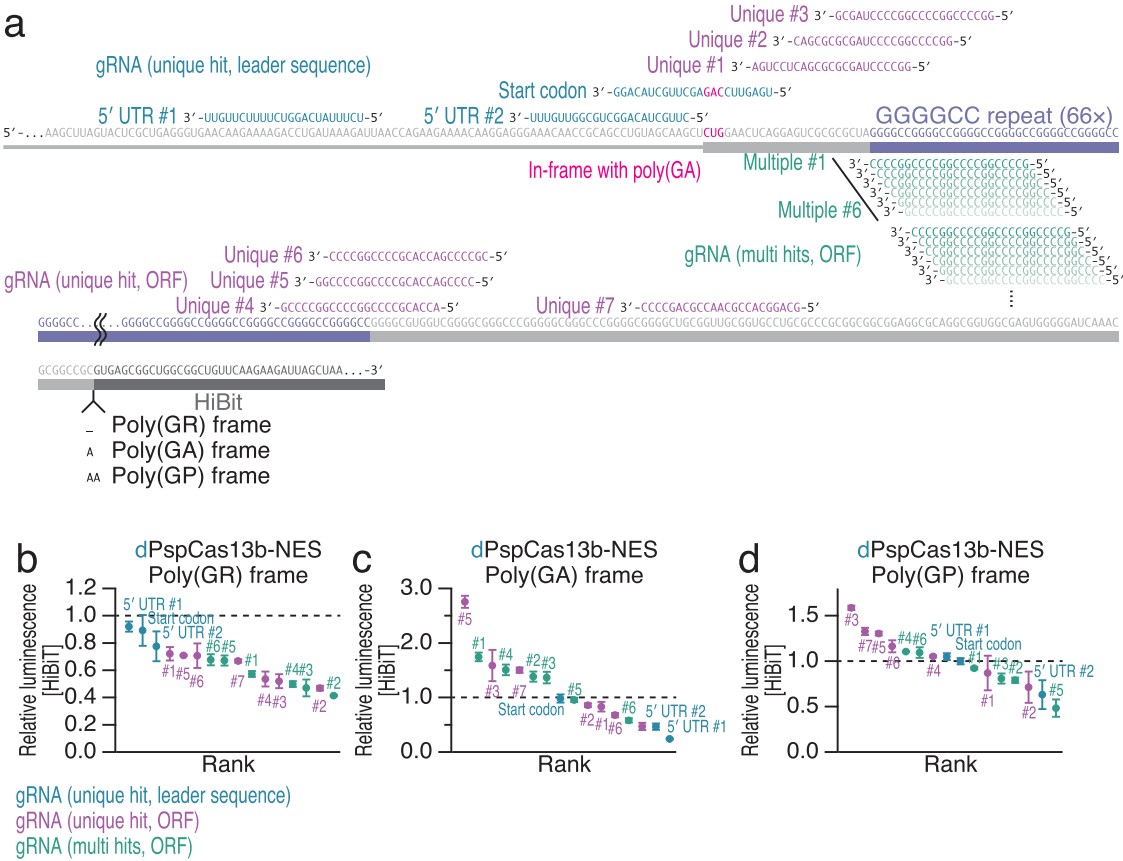

**Fig. 5 | CRISPRδ represses RAN translation. a** Schematic of the gRNA design for targeting various positions in GGGGCC repeat reporter mRNAs. **b**–**d** Relative HiBiT luminescence was calculated to quantify the repressive effects of dPspCas13b-NES using the gRNAs shown in (**a**). The mean (point) and s.d. (error) from three replicates are shown. The same data were used in Supplementary Fig. 5b–d.

Moreover, we also applied CRISPRδ to another endogenous gene, ANXA4, a calcium/phospholipid-binding protein, and detected translational repression via immunohistochemistry (Supplementary Fig. 6g–i). Overall, the results of the present study confirmed the potency of CRISPRδ for accessing genes at intermediate to high expression levels (Supplementary Fig. 6j).

Taken together, these data established CRISPRδ as a potent and highly specific tool for gene silencing through translational repression, independent of RNA degradation.

## Discussion

To overcome the collateral activity and toxicity stemming from the use of Cas13 as a gene silencing tool[21,47–57], the use of engineered Cas13 proteins[57], the controlled expression of the Cas13 protein[54], and the use of Cas13 variants with lower collateral activity[52] have been proposed. Despite the promising results, these approaches cannot completely prevent RNase activity toward bystander RNA species. In contrast, the CRISPRδ system presented in this study does not rely on RNA degradation and thus allows for a reduction in gene expression without the risk of induced collateral activity. Instead, CRISPRδ harnesses the high specificity of targeting by the dCas13-gRNA complex and the tight interaction of this complex with the target, resulting in steric hindrance of scanning ribosomes. This approach may be a sensitive means for general translational control since similar mechanisms are employed by natural translational regulators[151,152].

Recent reports have suggested that dCas13 could be a useful tool for modulating translation. In bacteria, targeting the Shine–Dalgarno (SD) sequence upstream of the start codon with dLwaCas13a or dRfxCas13d prevents hybridization of the anti-SD sequence in 16S rRNA and results in translational repression in vitro[153] and in vivo[154].

The mechanism of inhibition in the bacterial studies and this work differed due to the different principles of translation initiation in prokaryotes (internal recruitment of the small ribosome subunit via the SD/anti-SD interaction) and eukaryotes (scanning by the pre-initiation complex)[90]. However, all these works share the strategy of targeting a process before the formation of elongating ribosomes. Although those earlier studies lacked genome-wide investigations, this strategy may also have high specificity for gene silencing in bacteria, as highlighted in this work. In addition to inducing translational repression, the dCas13 approach may activate protein synthesis. Targeting IF3-fused dRfxCas13d to the 5′ UTR may augment translation in bacteria[154]. In mammals, conjugation of the SINEB2 repeat element, which activates protein synthesis from hybridized mRNA[155], to the gRNA of dRfxCa13d enhances translation[156]. In further studies, the applications of dCas13 will be expanded to translational control in a wide range of biological contexts across diverse organisms.

We expect translational repression mediated by CRISPRδ to have several advantages over preexisting methods. A wide array of studies have reported small but functional micropeptides encoded by long noncoding RNAs[157,158], which were originally expected to contain no protein-coding regions. Moreover, mRNAs have been reported to have protein coding-independent functions[159–161]. Thus, our method, which suppresses protein synthesis but leaves the target RNA intact, may be useful for delineating the roles of the RNA in protein production or other roles. Moreover, CRISPRδ is expected to circumvent the effects of the genetic compensation mechanism elicited by gene knockout by nonsense mutation[162,163], allowing straightforward interpretation of the gene expression–phenotype interaction. CRISPRδ represents a valuable step toward the development of a toolbox for reliable gene silencing and a distinct clarification of gene function.

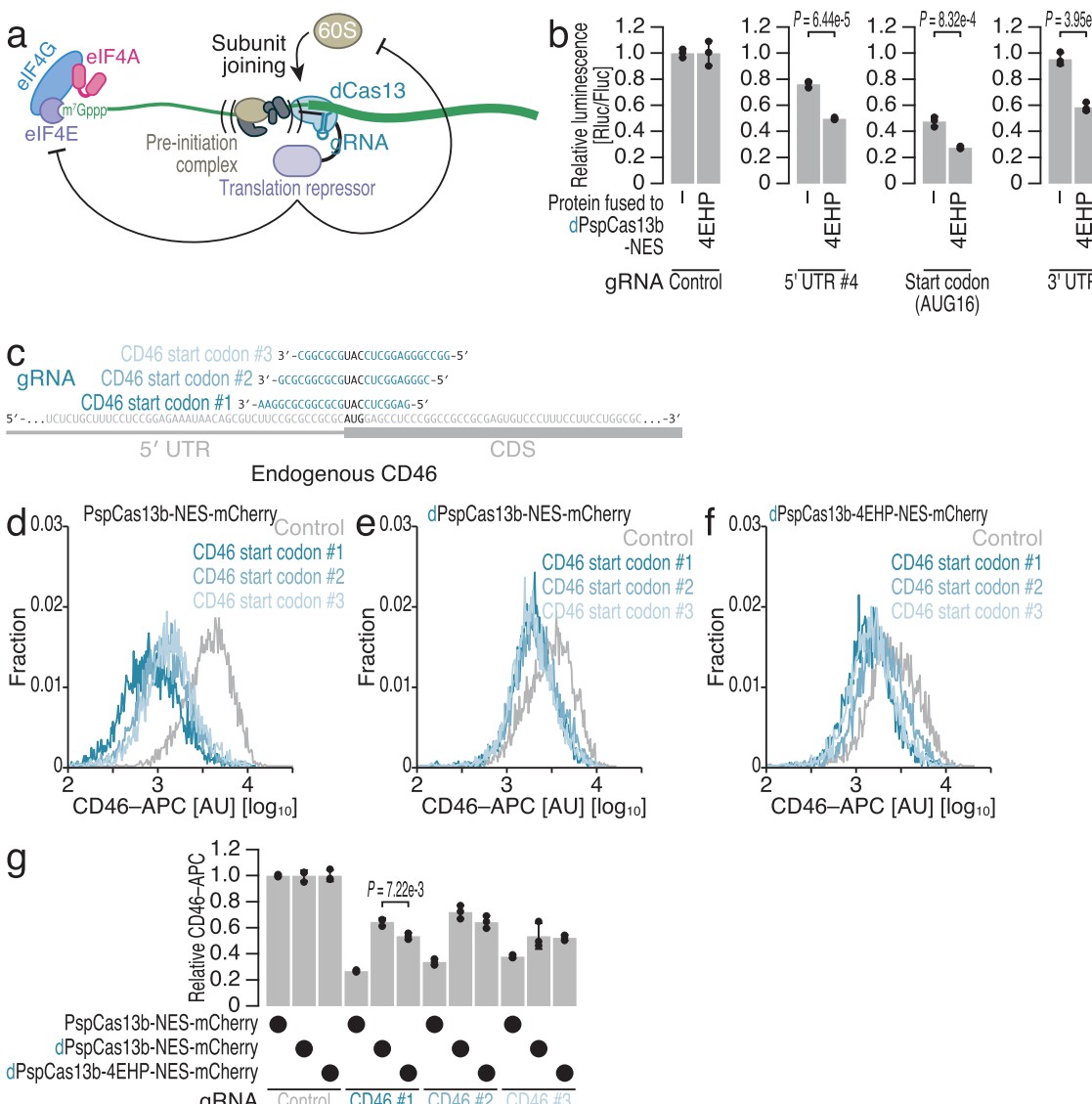

**Fig. 6 | Fusion of translational repressors to dPspCas13b enhances translational repression efficacy. a** Schematic showing the enhancement of dCas13-induced repression of mRNA translation by fusion with a translational repressor. Fused translational repressors provide an alternative mode of suppression. **b** Relative Rluc luminescence with respect to Fluc luminescence was calculated to quantify the repressive effects of dPspCas13b-NES and dPspCas13b-4EHP-NES using the gRNAs shown in Fig. 1e. **c** Schematic of the gRNA design for targeting the start codon of CD46 mRNA with PspCas13b-NES/dPspCas13b-NES/dPspCas13b-4EHP-NES. **d–f** Representative distribution of APC-labeled CD46 with the expression of the indicated PspCas13 variants and gRNAs. Typically, 4000–5000 cells that passed the expression threshold of mCherry fused to PspCas13 variants were considered. **g** Quantification of CD46 expression by FACS. The average values for the distributions (**d–f**) were used. In (**b**) and (**g**), the mean (gray bar), s.d. (black line), and individual replicates ($n = 3$, black points) are shown. In (**b**) and (**g**), $p$ values were determined by Student's $t$ test (two-tailed).

The relatively modest gene silencing is one of the caveats of the current CRISPRδ approach. Compared to RNAi-mediated knockdown, the efficiency and specificity of these methods are now in trade-offs: RNAi, high efficiency but moderate specificity; CRISPRδ, high specificity but moderate efficiency. Given that the stoichiometric abundance of the gRNA-Cas13 complex over target mRNAs is key for CRISPRδ, improving the low and stochastic expression of dCas13 proteins (Supplementary Figs. 1c, 3b) should be a promising avenue for increasing the efficiency of gene expression repression. Alternatively, direct introduction of an in vitro-assembled complex of recombinant dCas13 and gRNA to cells may be another approach, as described in earlier studies[34,43,45,62,64].

Although we focused mainly on NES-tagged dCas13 as a steric hindrance for scanning ribosomes in this study, NLS-tagged dCas13 may have extended the rationales for translational repression. We observed that ORF-targeting dPspCas13 with an NLS may have weak

potential for translation repression (Supplementary Fig. 1f), as if nuclear-enriched dCas13 (Supplementary Fig. 1i) may partially sequester the target mRNA into the nucleus and reduce accessibility to the translation machinery in the cytoplasm. Thus, NLS-tagged dPspCas13 may have dual functions in translational repression: steric hindrance for scanning by a fraction leaked in the cytoplasm and physical sequestration by a fraction enriched in the nucleus. Further study is required for the expansion of the CRISPRδ application and improvement of its performance.

## Methods
### Ethical statement
All animal care and experiments complied with the guidelines for animal experiments of the University of Tokyo and were approved by the animal research committee of the University of Tokyo.

## Cell cultures

The human embryonic kidney (HEK) 293 [American Type Culture Collection (ATCC), CRL-1573, female] line was cultured in high-glucose DMEM supplemented with GlutaMAX (Thermo Fisher Scientific) and 10% fetal bovine serum (FBS; Sigma–Aldrich) at 37 °C with 5% $CO_2$. Cell cultures were routinely tested for contamination with *Mycoplasma* spp. (e-Myco VALiD Mycoplasma PCR Detection Kit, iNtRON Biotechnology).

To establish a stable cell line expressing EGFP, HEK293 Flp-In T-REx cells (Thermo Fisher Scientific, R78007) were transfected with pcDNA5/FRT/TO-GFP (see below) and pOG44 (Thermo Fisher Scientific) using X-tremeGENE9 (Roche) and cultured in the presence of blasticidin S (InvivoGen) and hygromycin B (InvivoGen).

The human HeLa S3 (provided by RIKEN BioResource Research Center through the National Bio-Resource Project of the MEXT/AMED, Japan, RCB019, female) line was cultured in DMEM/Ham's F-12 (nacalai tesque) and 10% FBS (Sigma–Aldrich) on a shaker (EYELA) at 37 °C with 5% $CO_2$.

Mouse embryonic fibroblasts (MEFs) were isolated from E13 CD-1 mice (ICR, RRID:IMSR_CRL:22, Sankyo Labo Service Corporation). Embryos of either sex were eviscerated in HBSS and collected in a 50-ml tube. Small pieces of tissue were digested with 0.25% trypsin-EDTA (Fujifilm Wako Pure Chemical Corporation) for 20 min at 37 °C and plated for culture. MEFs were cultured in DMEM (Thermo Fisher Scientific) supplemented with 10% FBS, 1% GlutaMAX, and 1% penicillin–streptomycin (Thermo Fisher Scientific).

## Construction of pXR-LwaCas13a-NLS, LwaCas13a-NES, dLwaCas13a-NLS, and dLwaCas13a-NES

The DNA fragment encoding dLwaCas13a was PCR-amplified using pC035-dLwaCas13a-msfGFP (a gift from Feng Zhang; Addgene plasmid #91925; RRID: Addgene_91925)[17] as a template and inserted into pXR002: EF1a-dCasRx-2A-EGFP (a gift from Patrick Hsu; Addgene plasmid #109050; RRID: Addgene_109050)[19] between the C-terminal and N-terminal NLSs in the plasmid. To generate the plasmid that contained the NES, the DNA fragment encoding dLwaCas13a fused to a C-terminal NES sequence was PCR–amplified and inserted into pXR002: EF1a-dCasRx-2A-EGFP, eliminating the NLSs. To convert dLwaCas13a to the catalytically active WT LwaCas13a, the A474R, and A1046R substitutions were generated by site-directed mutagenesis.

## Construction of pXR-PguCas13-NLS, PguCas13-NES, dPguCas13-NLS, and dPguCas13-NES

The DNA fragment encoding PguCas13b was PCR-amplified, using pC0045-EF1a-PguCas13b-NES-HIV (a gift from Feng Zhang; Addgene plasmid #103861; RRID: Addgene_103861)[18] as a template and inserted into pXR002: EF1a-dCasRx-2A-EGFP[19] between the C-terminal and N-terminal NLSs in the plasmid. To generate the plasmid that contained the NES, the DNA fragment encoding PguCas13b fused to a C-terminal NES sequence was PCR-amplified and inserted into pXR002: EF1a-dCasRx-2A-EGFP, eliminating the NLSs. To convert PguCas13b to catalytically inactive dPguCas13b mutant, the H151A and H1121A substitutions were generated by site-directed mutagenesis.

## Construction of pXR-RfxCas13d-NLS, RfxCas13d-NES, dRfxCas13d-NLS, and dRfxCas13d-NES

pXR-dRfxCas13d-NLS was reported in the earlier study (originally named pXR002: EF1a-dCasRx-2A-EGFP)[19]. To generate the plasmid that contained the NES, the DNA fragment encoding dRfxCas13d fused to a C-terminal NES sequence was PCR-amplified and inserted into pXR002: EF1a-dCasRx-2A-EGFP, eliminating the NLSs. To convert dRfxCas13d to catalytically active WT RfxCas13d, the A239R, A244H, A858R, and A863H substitutions were generated by site-directed mutagenesis.

## Construction of pXR-PspCas13b-NLS, PspCas13b-NES, dPspCas13b-NLS, and dPspCas13b-NES

The DNA fragment encoding dPspCas13b was PCR-amplified, using pC0049-EF1a-dPSPCas13b-NES-HIV, H133A/H1058A (a gift from Feng Zhang; Addgene plasmid #103865; RRID: Addgene_103865)[18] as a template and inserted into pXR002: EF1a-dCasRx-2A-EGFP[19] between the C-terminal and N-terminal NLSs in the plasmid. To generate the plasmid that contained the NES, the DNA fragment encoding dPspCas13b fused to a C-terminal NES sequence was PCR-amplified and inserted into pXR002: EF1a-dCasRx-2A-EGFP, eliminating the NLSs. To convert dPspCas13b to catalytically active WT PspCas13b, the A133H and A1058H substitutions were generated by site-directed mutagenesis.

## Construction of pXR-dPspCas13b-PDCD4-NES, 4E-T-NES, eIF6-NES, 14-3-3σ-NES, PAIP2-NES, PELO-NES, and 4EHP-NES

The DNA fragments encoding the translational repressor proteins (PDCD4, 4E-T, eIF6, 14-3-3σ, PAIP2, PELO, or 4EHP) were synthesized (Eurofins Genomics), used as templates for PCR amplification, and fused to an XTEN10 linker[164,165]. The amplified fragments were then inserted between the dPspCas13b and NES sequences in pXR-dPspCas13b-NES.

## Construction of pCAGEN-PspCas13b-NES-mCherry, dPspCas13b-NES-mCherry, and PspCas13b-4EHP-NES-mCherry

The DNA fragments encoding PspCas13b-NES, dPspCas13b-NES, and dPspCas13b-4EHP-NES were PCR-amplified, using pXR-PspCas13b-NES, pXR-dPspCas13b-NES, and pXR-dPspCas13b-4EHP-NES, respectively, as templates and inserted into the pCAGEN vector (a gift from Dr. Yukihide Tomari at The University of Tokyo)[166], along with a PCR-amplified DNA fragment encoding mCherry.

## Construction of pXR-dPspCas13b-NLS-mCherry and pXR-dPspCas13b-NES-mCherry

To construct the pXR-dPspCas13b-NLS-mCherry plasmid, a PCR-amplified DNA fragment encoding mCherry was inserted into the pXR-dPspCas13b-NLS plasmid at the C-terminus of dPspCas13b-NLS. To construct the pXR-dPspCas13b-NES-mCherry plasmid, a PCR-amplified DNA fragment encoding mCherry was inserted into the pXR-dPspCas13b-NES plasmid at the C-terminus of dPspCas13b-NES.

## Construction of gRNA-expression plasmids

To construct the gRNA-expressing plasmids, the DR corresponding to each Cas13 ortholog, along with the respective spacer sequence, was inserted into the pC016-LwCas13a guide expression backbone with the U6 promoter (a gift from Feng Zhang; Addgene plasmid #91906; RRID: Addgene_91906). The gRNA expression plasmids used are listed in Supplementary Data 1.

## Construction of psiCHECK2-PTGES3

psiCHECK2-PTGES3, which contains the Rluc ORF fused to the PTGES3 5′ UTR, was a kind gift from Dr. Nicholas T. Ingolia at University of California, Berkeley[152].

## Construction of pCDNA5/FRT/TO-GFP

The PCR-amplified fragment encoding EGFP was inserted between the Hind III and BamH I sites in pcDNA5/FRT/TO (Thermo Fisher Scientific, V6520-20).

## Construction of psiCHECK2-HCV IRES

The psiCHECK2-HCV IRES, which contains the Rluc ORF fused to the HCV IRES, was a kind gift from Dr. Nicholas T. Ingolia at University of California, Berkeley[152].

## Construction of psiCHECK2-CrPV IRES-Nluc

The PCR-amplified DNA fragments encoding the CrPV IGR IRES and Nluc [using the pNL1.1 vector (Promega) as a template] were inserted into psiCHECK2 (Promega).

## Construction of pAG3-(GGGGCC)₆₆-HiBiT-GR frame, GA frame, and GP frame

A DNA fragment encoding the HiBiT tag (Promega) was inserted downstream of the $(GGGGCC)_{66}$ repeat in the plasmid pAG3-$(GGGGCC)_{66}$[107] [used as a poly(GR) frame construct], which was a kind gift from Dr. Leonard Petrucelli at the Mayo Clinic Jacksonville and Dr. Aaron D. Gitler at Stanford University. A single A nucleotide or AA dinucleotide was inserted upstream of the HiBiT tag to generate the GA frame and GP frame constructs.

## Construction of pAG3-empty

The DNA fragment encoding the $(GGGGCC)_{66}$ repeat was removed from the pAG3-$(GGGGCC)_{66}$ plasmid.

## Rluc and Fluc dual assay

Approximately $1 \times 10^5$ HEK293 cells were seeded in flat-bottom 24-well plates (Thermo Fisher Scientific or FALCON) and incubated overnight. For MEFs, $2.5 \times 10^4$ cells were seeded in 0.1% gelatin-coated flat-bottom 24-well plates (Thermo Fisher Scientific) and incubated overnight. The following day, the cells were cotransfected with the Cas13 variant-expressing plasmid (HEK293 cell, pXR series, 150 ng; MEFs, pCAGEN series, 150 ng), the gRNA-expressing plasmid (pC016 series, 300 ng), and the luciferase reporter plasmid (psiCHECK2-PTGES3, 5 ng; psiCHECK2-HCV IRES, 10 ng) using Lipofectamine 3000 (Thermo Fisher Scientific) according to the manufacturer's instructions. To assess the effects of sole gRNA and sole protein, the gRNA-expressing plasmid and the Cas13 variant-expressing plasmid were omitted. Forty-eight hours posttransfection, the cells were lysed with 100 µl of Passive Lysis Buffer (Promega). Then, 10 µl of lysate from each well was transferred to a 96-well flat-bottom white assay plate (Corning), and luminescence was measured by the Dual-Luciferase Reporter Assay System (Promega) and GloMax Navigator System (Promega).

For the HeLa S3 suspension cells, ~$3 \times 10^6$ HeLa S3 cells were seeded in Erlenmeyer flasks (Corning) in 15 ml of DMEM/Ham's F-12 (nacalai tesque) supplemented with 10% FBS (Sigma–Aldrich) and incubated overnight on a shaker (EYELA). The following day, the cells were cotransfected with the Cas13 variant-expressing plasmid (pCAGEN series, 4.5 µg), the gRNA-expressing plasmid (pC016 series, 9 µg), and the luciferase reporter plasmid (psiCHECK2-PTGES3, 150 ng) using Lipofectamine 2000 (Thermo Fisher Scientific) according to the manufacturer's instructions. Forty-eight hours posttransfection, the cells were collected by centrifugation at $800 \times g$ for 3 min. The cells were washed with PBS (nacalai tesque) and centrifuged at $800 \times g$ for 3 min. The supernatant was removed, and the cells were lysed with 300 µl of Passive Lysis Buffer (Promega). Then, 10 µl of lysate was analyzed as described above.

For confirmation of IRES-mediated translation initiation, 10 ng of psiCHECK2-PTGES3 or psiCHECK2-HCV IRES was used. Hippuristanol (1 µM with 0.1% DMSO as a stock solvent) (a gift from Dr. Junichi Tanaka at the University of the Ryukyus) or DMSO (0.1%) was added to the cells 10 h before lysis.

## Nluc assay

The template DNA fragments for in vitro transcription were PCR-amplified using psiCHECK-CrPV IRES-NLuc with primers (5′-TAATACGACTCACTATAGG-3′ and 5′-CACACAAAAAACCAACACACAG-3′). The A-capped reporter RNA was transcribed with a T7-Scribe Standard RNA IVT Kit (CELLSCRIPT) in the presence of 6 mM G(5′)ppp(5′)A RNA Cap Structure Analog (Jena Bioscience), 7.5 mM ATP, 7.5 mM CTP, 7.5 mM UTP, and 1.5 mM GTP. Then, the mRNA was polyadenylated with an A-Plus Poly(A) Polymerase Tailing Kit (CELLSCRIPT).

Approximately $1 \times 10^5$ HEK293 cells were seeded in flat-bottom 24-well plates (Thermo Fisher Scientific) and incubated overnight. The following day, the cells were cotransfected with the Cas13 variant-expressing plasmid (pCAGEN-dPspCas13b-NES-mCherry, 150 ng) and the gRNA-expressing plasmid (pC016 series, 300 ng) using Lipofectamine 3000. Forty-eight hours posttransfection, the cells were transfected with the reporter mRNA (50 ng) using the TransIT-mRNA Transfection Kit (Mirus). After incubating for 16 h, the cell lysis was performed as described above. Luciferase activity was measured by the Nano-Glo Luciferase Assay System (Promega) and GloMax Navigator System (Promega).

For preparation of the m⁷G-capped mRNA, the DNA fragments were PCR-amplified from psiCHECK-CrPV IRES-NLuc with primers (5′-TAATACGACTCACTATAGGCCATGGAAGACGCCAAAAACATAAA-3′ and 5′-CACACAAAAAACCAACACACAG-3′). The reporter RNA was transcribed, capped, and polyadenylated using T7-Scribe Standard RNA IVT Kit (CELLSCRIPT), ScriptCap m⁷G Capping System (CELLSCRIPT), ScriptCap 2′-O-Methyltransferase Kit (CELLSCRIPT), and A-Plus Poly(A) Polymerase Tailing Kit (CELLSCRIPT). The cells were transfected with 50 ng of mRNAs and treated with hippuristanol (1 µM with 1.0% DMSO as a stock solvent) or DMSO (1.0%) at 12 h before lysis.

## HiBiT assay

Approximately $1 \times 10^5$ HEK293 cells were seeded in flat-bottom 24-well plates (Thermo Fisher Scientific) and incubated overnight. The following day, the cells were cotransfected with the Cas13 variant-expressing plasmid (pCAGEN series, 150 ng), the gRNA-expressing plasmid (pC016 series, 300 ng), and the HiBiT reporter plasmid [pAG3-$(GGGGCC)_{66}$-HiBiT-GR frame and GP frame, 400 ng; pAG3-$(GGGGCC)_{66}$-HiBiT-GA frame, 75 ng] using TransIT-293 (Mirus) according to the manufacturer's instructions. Forty-eight hours posttransfection, 400 µl of the medium was removed from each well, and 100 µl of HiBiT Lytic Reagent (Nano Glo HiBiT Lytic Detection System, Promega) was added to the remaining 100 µl of the medium. After incubating for 10 min on an orbital shaker, the total lysate was transferred to a 96-well flat-bottom white microplate (Greiner Bio-One or Corning), and luminescence was measured by the GloMax Navigator System (Promega).

## Data analysis of luciferase assays

The average of the background luminescence signals (Rluc, Fluc, Nluc, and HiBiT) from triplicate wells of nontransfected cells was subtracted from the luminescence values of the experimental wells. The luminescence values were normalized to the average of the corresponding triplicate control wells.

## Labeling of nascent peptides by O-propargyl-puromycin (OP-puro)

Approximately $1 \times 10^5$ HEK293 cells were seeded in flat-bottom 24-well plates (Thermo Fisher Scientific) and incubated overnight. The following day, the cells were cotransfected with the Cas13 variant-expressing plasmid (pXR series, 150 ng), the gRNA-expressing plasmid (pC016-gPsp-Control, 300 ng), and the luciferase reporter plasmid (psiCHECK2-PTGES3, 5 ng) using Lipofectamine 3000 according to the manufacturer's instructions. Forty-eight hours posttransfection, the cells were treated with 20 µM OP-puro (Jene Bioscience) for 30 min at 37 °C with 5% $CO_2$, washed with PBS, lysed in OP-puro lysis buffer (20 mM Tris-HCl pH 7.5, 150 mM NaCl, 5 mM $MgCl_2$, and 1% Triton X-100), and then clarified via centrifugation at $20,000 \times g$ for 10 min at 4 °C. The lysates were incubated with 1 µM IRDye800CW Azide (LI-COR Bioscience) for 30 min at 25 °C using a Click-it cell reaction kit (Thermo Fisher Scientific) according to the manufacturer's instructions and subsequently subjected to SDS–PAGE. Images from the gels

were acquired and quantified using an ODYSSEY CLx (LI-COR Biosciences). Total protein staining was subsequently performed on the gels using GelCode Blue Safe Protein Stain (Thermo Fisher Scientific). The proteins were quantified using an ODYSSEY CLx (LI-COR Biosciences) and used for the normalization of OP-puro-labeled nascent protein signals.

### RT–qPCR

Approximately $4 \times 10^5$ cells were seeded in flat-bottom six-well plates (Thermo Fisher Scientific) and incubated overnight. The following day, the cells were cotransfected with the Cas13 variant-expressing plasmid (pXR series, 600 ng), the gRNA-expressing plasmid (pC016 series, 1200 ng), and the luciferase reporter plasmid (psiCHECK2-PTGES3, 20 ng) using Lipofectamine 3000 (Thermo Fisher Scientific) according to the manufacturer's instructions. RNA was harvested 48 h post-transfection using TRIzol reagent (Thermo Fisher Scientific) and a Direct-zol RNA MicroPrep Kit (Zymo Research) according to the manufacturer's instructions. Then, the RNA was treated with DNase (TURBO DNA-free Kit, Thermo Fisher Scientific), purified with RNA-Clean XP beads (Beckman Coulter), and reverse-transcribed using ReverTra Ace qPCR RT Master Mix (TOYOBO). qPCR was performed on a thermal cycler (TaKaRa or Bio-Rad) using TB Green Premix Ex TaqTM II (TaKaRa) or iTaq Universal SYBR Green Supermix (Bio-Rad) with the following primers: Rluc, 5′-TCGTCCATGCTGAGAGTGTC-3′ and 5′-CTAACCTCGCCCTTCTCCTT-3′; Fluc, 5′-TTCGCTAAGAGCACCCTGAT-3′ and 5′-GTAATCAGAATGGCGCTGGT-3′; CD46, 5′-GGGATCCCCCAGTTCCAAAG-3′ and 5′-GCACTGGACGCTGGAGATTT-3′; HiBiT, 5′-TAACCAGAAGAAAACAAGGAGGGA-3′ and 5′-TCAGGAGTCGCGCGC-3′; and β-Actin, 5′-AGAGCTACGAGCTGCCTGAC-3′ and 5′-AGCACTGTGTTGGCGTACAG-3′.

### Western blot

Anti-HA (Roche, 3F10, 11867423001, 1:1000), anti-GFP (Cell Signaling Technology, 2956, 1:1000), and anti-β-Actin (LI-COR Biosciences, 926-42212, 1:1000) were used as primary antibodies. As secondary antibodies, anti-rat IgG IRDye 800CW (LI-COR Biosciences, 926-32219, 1:10,000), anti-mouse IgG IRDye 680CW (LI-COR Biosciences, 926-68070, 1:10,000), anti-mouse IgG IRDye 800CW (LI-COR Biosciences, 925-32210, 1:10,000), and anti-rabbit IgG IRDye680CW (LI-COR Biosciences, 925-68071, 1:10,000) were used. Images were acquired with an Odyssey CLx (LI-COR Biosciences) and Image Studio (LI-COR Biosciences, version 5.2). Antibodies were validated by the manufacturers.

### Microscopic analysis

HEK293 cells were seeded onto poly-D-lysine-coated dishes (35-mm glass bottom, MatTek) and incubated overnight. The following day, the cells were cotransfected with the Cas13 variant-expressing plasmid (pXR-dPspCas13b-NLS-mCherry or pXR-dPspCas13b-NES-mCherry, 300 ng), the gRNA-expressing plasmid (pC016 series, 600 ng), and the luciferase reporter plasmid (psiCHECK2-PTGES3, 10 ng) using TransIT-293 (Mirus) according to the manufacturer's instructions. The cells were incubated with 0.5 μg/ml Hoechst 33342 (nacalai tesque) for 15 min and observed under an LSM 710 (Carl Zeiss) confocal microscope with an Objective Plan-Apochromat 63×/1.4 Oil DIC M27 (Carl Zeiss).

### FACS for assessment of EGFP expression

Approximately $4 \times 10^6$ HEK293 Flp-In T-Rex cells with EGFP integrants were seeded in 15-cm dishes (Thermo Fisher Scientific) and incubated overnight. The following day, the cells were cotransfected with the Cas13 variant-expressing plasmid (pCAGEN-PspCas13b-NES-mCherry or pCAGEN-dPspCas13b-NES-mCherry, 8 μg) and the gRNA-expressing plasmid (pC016-gPsp-Control or pC016-gPsp-EGFP start codon, 12 μg) using TransIT-293 (Mirus) according to the manufacturer's instructions. The cells were then incubated for 48 h prior to FACS. To induce

EGFP expression, the cells were treated with tetracycline (1 μg/ml) for 24 h prior to FACS.

The cells were washed with PBS (nacalai tesque) and trypsinized (Thermo Fisher Scientific). Then, the cells were resuspended in high-glucose DMEM supplemented with GlutaMAX (Thermo Fisher Scientific) and 10% FBS (Sigma–Aldrich) and centrifuged at $800 \times g$ for 3 min. After the supernatant was removed, the cells were washed with PBS and centrifuged again at $800 \times g$ for 3 min. The cell pellet was resuspended in 1.5 ml of PBS. After passing through a 35-μm mesh filter (FALCON), the cells were analyzed via a FACSAria II Special Order system (BD) with FACSDiva ver. 6.1.3 software (BD) to measure EGFP and mCherry expression. The data from $1 \times 10^4$ cells were used for analysis. The gating strategy can be found in Supplementary Fig. 7.

### FACS for assessment of CD46 expression

Approximately $4 \times 10^5$ HEK293 cells were seeded in flat-bottom 6-well plates (Thermo Fisher Scientific) and incubated overnight. The following day, the cells were cotransfected with the Cas13 variant-expressing plasmid (pCAGEN-PspCas13b-NES-mCherry, pCAGEN-dPspCas13b-NES-mCherry, or pCAGEN-dPspCas13b-4EHP-NES-mCherry, 800 ng) and the gRNA-expressing plasmid (pC016-gPsp-Control, pC016-gPsp-CD46 #1, pC016-gPsp-CD46 #2, or pC016-gPsp-CD46 #3, 1200 ng) using TransIT 293 (Mirus) according to the manufacturer's instructions. The cells were then incubated for 48 h prior to FACS.

The cells were washed with PBS (nacalai tesque) and trypsinized (Thermo Fisher Scientific). Then, the cells were resuspended in high-glucose DMEM supplemented with GlutaMAX (Thermo Fisher Scientific) and 10% FBS (Sigma–Aldrich) and centrifuged at $800 \times g$ for 3 min. After the supernatant was removed, the cells were washed with FACS Buffer [2% FBS and 0.1% NaN$_3$ (nacalai tesque) in PBS (nacalai tesque)] and centrifuged at $800 \times g$ for 3 min. The cells were resuspended in 10 μl of FACS Buffer and incubated with 1 μl of allophycocyanin (APC)-conjugated anti-CD46 antibody (BioLegend, clone TRA-2–10, 352405, validated by the manufacturer) for 30 min at 4 °C with rotation. Then, the cells were washed twice with FACS Buffer and resuspended in 1 ml of FACS Buffer. After passing through a 35-μm mesh filter (FALCON), the cells were analyzed via a FACSAria II Special Order system (BD) with FACSDiva 8.0.2 software (BD) to measure APC signals and mCherry expression. The background signal for mCherry was measured in cells not transfected with an mCherry-fused Cas13 variant expression plasmid and used for the identification of mCherry-positive cells (a value of $10^3$ or higher). The gating strategy can be found in Supplementary Fig. 7.

### FACS for assessment of ANXA4 expression

Approximately $4 \times 10^6$ HEK293 cells were seeded in 10-cm dishes (Thermo Fisher Scientific) and incubated overnight. The following day, the cells were cotransfected with the Cas13 variant-expressing plasmid (pCAGEN-dPspCas13b-NES-mCherry, 4 μg) and the gRNA-expressing plasmid (pC016-gPsp-Control or pC016-gPsp-ANXA4#1, 6 μg) using TransIT 293 (Mirus) according to the manufacturer's instructions. The cells were then incubated for 48 h prior to FACS.

The cells were washed with PBS (nacalai tesque) and trypsinized (Thermo Fisher Scientific). Then, the cells were resuspended in high-glucose DMEM supplemented with GlutaMAX (Thermo Fisher Scientific) and 10% FBS (Sigma–Aldrich) and centrifuged at $300 \times g$ for 3 min. After the supernatant was removed, the cells were washed with PBS (nacalai tesque). For fixation, the cells were treated with 4% PFA (nacalai tesque) and incubated for 15 min on ice. The cells were then centrifuged at $300 \times g$ for 3 min, after which the supernatant was removed. For permeabilization, the cells were incubated with 0.1% (v/v) Triton X-100 in PBS for 5 min at room temperature. The cells were resuspended in 10 μl of Intercept (TBS) Blocking Buffer (LI-COR) with 1 μl of anti-ANXA4 antibody (Abcam, ab153883, validated

by the manufacturer) for 30 min at 4 °C with rotation. Following centrifugation at $1000 \times g$ for 3 min, the cells were resuspended in 10 µl of Intercept (TBS) Blocking Buffer with anti-rabbit Alexa Fluor 488 secondary antibody (Thermo Fisher Scientific, SA5-10323, 1:1000, validated by the manufacturer) and incubated for 1 h at 4 °C with rotation. Then, the cells were washed twice with PBS and resuspended in 500 µl of PBS. After passing through a 35-µm mesh filter (FALCON), the cells were analyzed via a FACSAria II Special Order system (BD) with FACSDiva 8.0.2 software (BD) to measure Venus-A signals and mCherry expression. The background signal for mCherry was measured in cells not transfected with an mCherry-fused Cas13 variant expression plasmid and used for the identification of mCherry-positive cells (a value of $10^2$ or higher). The gating strategy can be found in Supplementary Fig. 7.

The specificity of the anti-ANXA4 antibody used for immunohistochemistry was further validated by siRNA-mediated knockdown (Dharmacon, L-010742-00-0005).

### Library preparation of ribosome profiling and RNA-Seq

The preparatin of ribosome profiling library followed the method reported previously[167] with modifications. Cell seeding, transfection, and sorting were performed as described in the "*FACS for assessment of EGFP expression*". Cycloheximide (100 µg/ml, Sigma–Aldrich) was added to PBS, trypsin solution, and DMEM to halt ribosome movement along mRNAs in the harvested cells. Then, the cells were subjected to FACS as described in the "*FACS for assessment of EGFP expression*". Based on the result of cells without mCherry-containing plasmids, we collected cells with a value of $10^2$ or higher as mCherry-positive cells. The sorted $1 \times 10^6$ cells were pelleted by centrifugation at $800 \times g$ for 3 min and subsequently lysed in lysis buffer (20 mM Tris-HCl pH 7.5, 150 mM NaCl, 5 mM MgCl$_2$, 1 mM DTT, 1% Triton X-100, 100 µg/ml cycloheximide, and 100 µg/ml chloramphenicol). The lysates were treated with Turbo DNase (Thermo Fisher Scientific) for 10 min at 4 °C and then clarified by removal of cellular debris by centrifugation at $20{,}000 \times g$ for 10 min at 4 °C. The total RNA concentration in each lysate was measured using a Qubit RNA BR Assay Kit (Thermo Fisher Scientific). Lysates containing 3 µg of total RNA were treated with 20 U of RNase I (Lucigen) for 45 min at 25 °C and then subjected to sucrose cushion ultracentrifugation at $542000 \times g$ for 1 h at 4 °C with an Optima MAX-TL ultracentrifuge and a TLA-110 rotor (Beckman Coulter). The RNA fragments in the resulting pellet were recovered using TRIzol reagent (Thermo Fisher Scientific) and a Direct-zol RNA MicroPrep Kit (Zymo Research) and subjected to denaturing polyacrylamide gel electrophoresis (PAGE). RNA fragments with lengths ranging between 17 and 34 nt were excised from the gel. After purification, the RNA fragments were dephosphorylated by incubation with T4 polynucleotide kinase (New England Biolabs) for 1 h at 37 °C and ligated to preadenylated linker oligonucleotides (5'-App NNNNNIIIIIAGATCGGAAGAGCACACGTCTGAA-ddC-3', where App indicates preadenylation, ddC indicates 2′,3′-dideoxycytidine, Ns indicates a random sequence for the unique molecular index, and Is indicates the index sequence used for multiplexing) by incubation with T4 RNA Ligase 2, truncated KQ (New England Biolabs) for 3 h at 22 °C in 17.5% PEG-8000. The linker-ligated RNA was subjected to denaturing PAGE and gel extraction. rRNA was removed from the purified RNA using a Ribo-Zero Gold rRNA Removal Kit (Human/Mouse/Rat) accompanied by a TruSeq Stranded Total RNA Kit (Illumina). RNA was then reverse-transcribed for 30 min at 50 °C using ProtoScript II (New England Biolabs) with the primer 5′-Phos-NNAGATCGGAAG AGCGTCGTGTAGGGAAAGAG-iSp18-GTGACTGGAGTTCAGACGTGT GCTC-3′, where Phos indicates 5′ monophosphate and iSp18 indicates an 18-atom hexaethylene glycol spacer. The template RNA was hydrolyzed with 1 M NaOH and subjected to denaturing PAGE and gel extraction. The purified cDNA was then circularized using CircLigase II ssDNA ligase (Epicenter) for 1 h at 60 °C and was subsequently PCR-

amplified using the primers 5′-AATGATACGGCGACCACCGAGATCTAC ACTCTTTCCCTACACGACGCTC-3′ and 5′-CAAGCAGAAGACGGCA-TACGAGATJJJJJJGTGACTGGAGTTCAGACGTGTG-3′ (where Js indicates the 6-mer index sequence for Illumina sequencing).

For RNA-Seq library preparation, total RNA was extracted from the same lysate used for ribosome profiling using TRIzol LS reagent (Thermo Fisher Scientific) and a Direct-zol RNA MiniPrep Kit (Zymo Research). The library was prepared using a TruSeq RNA Library Preparation Kit v2 (Illumina) according to the manufacturer's instructions.

DNA libraries were sequenced on the HiSeq X (Illumina) platform with the paired-end 150-nt option.

### Data analysis of ribosome profiling and RNA-Seq

Sequencing data analysis followed the reported pipelines[168,169] with the deposited codes (Zenodo, https://zenodo.org/records/7477706). To perform read quality filtering and adapter trimming, we used fastp (ver. 0.21.0)[170]. STAR (ver. 2.7.0a)[171] was used to remove reads aligned to noncoding RNAs. The remaining reads were mapped to the hg38 human genome by STAR and annotated with the GENCODE Human Release 32 reference obtained via the UCSC Genome Browser (https://genome.ucsc.edu/index.html).

We determined the offset of the A site in the reads according to the metagene analysis conducted with a custom script (https://github.com/ingolia-lab/RiboSeq) to be 15 for 25–33-nt reads. For RNA-Seq, we used an offset of 15 for all mRNA fragments. To calculate the changes in RNA abundance and translation efficiency, which were calculated as ribosome profiling counts normalized to RNA-Seq counts, we used the DESeq2 package (ver. 1.32.0)[172] with R software (ver. 4.1.1) in the RStudio interface (ver. 2021.09.0 + 351). We calculated the significance with the likelihood ratio test in a generalized linear model. The reads that corresponded to the first five codons and the last five codons in the ORF were excluded from our analyses.

### Cell viability assay

Approximately $2 \times 10^3$ HEK293 cells were seeded in flat-bottom 96-well plates (Greiner) and incubated overnight. For Supplementary Fig. 5a, the following day, the cells were cotransfected with the Cas13 variant-expressing plasmid (pCAGEN series, 30 ng), the gRNA-expressing plasmid (pC016 series, 60 ng), and the HiBiT reporter plasmid [pAG3-(GGGGCC)$_{66}$-HiBiT-GR frame, 80 ng]; alternatively, the cells were transfected with either an empty reporter plasmid vector (pAG3-empty, 80 ng) or the HiBiT reporter plasmid [pAG3-(GGGGCC)$_{66}$-HiBiT-GR frame, 80 ng] alone. For Supplementary Fig. 6e, the following day, the cells were transfected with the Cas13 variant-expressing plasmid (pXR series, 30 ng), the gRNA-expressing plasmid (pC016-gPsp-Control, 60 ng), and the luciferase reporter plasmid (psiCHECK2-PTGES3, 1 ng). Both transfections were performed using TransIT-293 (Mirus) according to the manufacturer's instructions. Cell viability was assessed at 48 h posttransfection using RealTime-Glo MT Cell Viability Assay reagent (Promega) according to the manufacturer's instructions. The luminescence was detected by a GloMax Navigator.

### Reporting summary

Further information on research design is available in the Nature Portfolio Reporting Summary linked to this article.

## Data availability

The ribosome profiling and RNA-Seq data (GSE232383) obtained in this study were deposited in the National Center for Biotechnology Information (NCBI) database. Gene annotation (GENCODE Human Release 32 reference) was obtained via the UCSC Genome Browser (https://genome.ucsc.edu/index.html). Source data are provided with this paper.

## Code availability

For ribosome profiling and RNA-Seq data analysis, we used the deposited codes (Zenodo 7477706)[169] and amended them accordingly.

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

## Acknowledgements

We are grateful to all the members of the Iwasaki laboratory for their constructive discussions and technical help. We thank Dr. Aaron D. Gitler for the critical reading of the manuscript. We also thank the Support Unit for Bio-Material Analysis, RIKEN CBS Research Resources Division, for Sanger sequencing and FACS analysis. Computation was supported by the HOKUSAI SailingShip supercomputer facility at RIKEN. We also thank Dr. Junichi Tanaka for sharing hippuristanol with us. pCAGEN vector was a kind gift from Dr. Yukihide Tomari. pAG3-(GGGGCC)₆₆ vector was a kind gift from Dr. Leonard Petrucelli and Dr. Aaron D. Gitler. psiCHECK2-PTGES3/HCV IRES vectors were kind gifts from Dr. Nicholas T. Ingolia. This work was supported by the Ministry of Education, Culture, Sports, Science and Technology (MEXT) (JP20H05784 to S.I.; JP21H05278 to H.T.; JP20H05786 to Y.I.; JP21H05734 and JP23H04268 to Y.S.); the Japan Society for the Promotion of Science (JSPS) (JP23H02415 to S.I.; JP23KJ2175 to N.K.; JP20K07016 to. H.T.; JP21K15023 and JP23K05648 to Y.S.); the Japan Agency for Medical Research and Development (AMED) (JP20gm1410001 to S.I., H.T., and Y.I.; JP23gm6910005 to Y.S.); RIKEN (Pioneering Projects Biology of Intracellular Environments to S.I. and Y.S.); and the Exploratory Research Center on Life and Living Systems (ExCELLS) (23EX601 to Y.S.). A.A was an International Program Associate of RIKEN. N.K. was a recipient of the JSPS Research Fellows (PD).

## Author contributions

A.A., H.T., Y.S., and S.I. conceptualized the experiments. A.A., Y.S., and S.I. developed the methodology. A.A., N.K., S.Y.A.C., Y.S., and S.I. performed formal analysis. A.A., N.K., S.Y.A.C., and Y.S. conducted the experiments. A.A. and S.I. wrote the original draft. All the authors reviewed and edited the manuscript. A.A., Y.S., and S.I. visualized and presented the data. H.T., Y.I., Y.S., and S.I. supervised the experiments and analyses. S.I. managed the project. N.K., H.T., Y.I., Y.S., and S.I. acquired the financial support related to this manuscript.

## Competing interests

The authors declare no competing interests.
