## [Peer Review File · Nature Communications]

Reviewers' Comments:

Reviewer #1:

Remarks to the Author:

Apostolopoulos et al. describe an innovation use of deactivated Cas13, termed CRISPRdelta, to inhibit translation in mammalian cells, but not affect mRNA levels. The authors provide very strong data to support their conclusions and should be commended for the overall systematic dissection they performed. The authors first test multiple different Cas13 homologs to identify that that dPspCas13-NES is most affective and then proceed to testing and optimizing the gRNA that targets the start codon of luciferase. The rules learned here did generally apply to other mRNAs, but was less effective against the GGGGCC repeat and RAN translation; however, this is perhaps expected with much of the mechanism of GGGGCC RAN translation still unknown and that a single repeat mRNA is translated in multiple reading frames. In nearly every experiment, the authors also include the active PspCas13b as a control to demonstrate that the gRNAs in theory do function and can recognize the target. This was important because it gives validity to the read out of the translational repression assays with dPspCas13b and the many gRNAs tested. The ribosome profiling in GFP expressing stable lines demonstrates that only WT PspCas13, and not dPspCas13, collateral effects. The manuscript ends on the ability of CRISPRdelta to target endogenous mRNAs, which was extremely important for the authors to demonstrate. In total, the manuscript is focused, well-written, novel, and would be of general interest.

I have four major critiques for the authors:

1) It is clearly demonstrated that CRISPRdelta is effective against reporter mRNAs during transient co-transfection in HEK293 cells (~5-fold reduction in Luc activity). The experiments testing transcripts originating from a chromosome (e.g, the stable GFP line in Figure 3 or CD46 in Figure 6) appear to be less robust (e.g., ~2-fold reduction in the GFP in the stable line in cells that got the most mCherry-tagged dCas13), especially in the light that RNAi with commercial siRNAs produce nearly 100% knockdown of the target protein. I cannot tell if the assay in Figure 6 used the entire cell population or just the highly transfected pool like in Figure 3. This certainly may be due to the transfection/expression pattern one often gets from transient plasmid transfection.

My advice is two-fold: i) Unlike plasmid transfections, mRNA transfections with Lipofectamine MessengerMAX, at least in our hands, gives robust, even, and complete transfection. My thought is using this delivery method would provide a more dramatic effect when assayed by Western or FACS; ii) using the more efficacious fused translational repressor as in Figure 6 in addition to the point directly above could provide more robust repression for endogenous targets.

With that being said, this critique would be voided if the authors can provide a Western blot showing >5-fold reduction for CD46 in the total cell population (related #2 below).

2) Since most users of CRISPRdelta would realistically use Westerns (and not ribosome profiling or FACS) to confirm translational repression and probably not easily be able to use FACS in their desired experiments, it would be highly valuable to demonstrate reduction of a targeted endogenous protein by Western blot. CD46 would suffice of course. We have found that siRNA knockdown of GAPDH is doable after 48 hrs; so that is another option. I believe any endogenous protein would suffice and prove the validity of the technique to most future users.

3) I apologize if I missed it, it appears all experiments have been performed in HEK293 cells. It is important to provide evidence that CRISPRdelta would be nearly as functional in another cell type. To be clear, this is NOT a request to repeat everything in a different cell type. I recommend a robust and well controlled experiment shown in Figure 1E or Figure 6B.

4) It would be very strong to demonstrate mRNA levels are not changing in Figure 6 when using the fused translational repression version, for both the luciferase and CD46. The OPP measurement was great and does provide evidence to the readers that global translation is not affected.

Minor critiques:

5) The authors mention at multiple locations about the tight binding of Cas13 to mRNA, but they make the readers look into previous literature for what this really is. Can the authors include

reported affinity measurements in the text?

6) In Figure 1B, are all Cas13 homologs equally expressed? This could be an explanation to why only some of the WT homologs are functional. This is not a major point since the main focus of the paper is to demonstrate that at least a few can be used to repress translation.

7) It may clearer to keep the "translational repression" nomenclature instead of "gene knockdown" when referring to the CRISPRdelta since nearly the entire field always uses knockdown when describing RNAi.

8) Rationale is not entirely laid out for testing the NES and NLS versions in Figure 1C. Nor is it demonstrated, even on dPspCas13b, they are functional on control transfections with no gRNA.

9) In the methods for CD46 expression via FACS, it is not specified what cell type was used. I assume HEK293 cells again.

Reviewer #2:

Remarks to the Author:

In the manuscript titled "CRISPR δ : dCas13-mediated translational repression for accurate gene silencing in mammalian cells", the authors developed a dCas13-based method to inhibit the protein production from a specific target mRNA without affecting mRNA abundance. This approach also avoids the collateral activity associated with wild type Cas13-based RNA knockdown. This study not only provides a new tool for loss-of-function studies of protein-coding genes, but also opened the door for studying protein-independent function of coding mRNAs. The method represents an important new application of RNA-targeting CRISPR systems in mammalian cells. The research is novel considering the differences between bacterial and mammalian expression systems.

Overall, the experiments were carefully designed and carried out by an experienced group. The authors systematically optimized several parameters of the system, including Cas13 effectors, subcellular localization of effectors (NLS vs NES), gRNA length, target position, etc. The system has been tested with luciferase reporters (Cap-dependent and IRES-dependent), a GFP reporter, a disease-relevant reporter (G4C2 repeat), and an endogenous target (CD46). The specificity of the approach is supported by compelling RNA-seq and ribosome-profiling data.

I strongly support the publication of this study, if the authors can address a few concerns detailed below.

Major comments:

Mechanism of action: while steric hindrance of translation initiation at the start codon remains the most likely explanation for the observed decrease in protein activity/abundance, several observations are not consistent with this model and will need clarifications or control experiments to rule out alternative mechanisms. For example, there is a negligible difference between NLS- and NES-containing dCas13b at the start codon (Fig. 1C). One would expect an NLS-containing dCas13b to be unable to influence translation. One possibility is that NLS-dCas13b is loaded in the nucleus and then exported together with the mRNA to the cytoplasm, which then inhibits translation initiation. Or, NLS-dCas13b may inhibit translation via a different mechanism, e.g., sequestering target mRNAs in the nucleus and prevents translation. A discussion on these possibilities is recommended. Another alternative model that needs to be excluded is that, dCas13b may not be required for target inhibition, i.e., the gRNA itself is sufficient to induce inhibition via steric hindrance, RNA editing, or RNAi-like effect. The inefficiency of other Cas13 proteins does not rule out this possibility, as the scaffold regions are different for each subtype of Cas13 system. A control experiment in which dPspCas13b is omitted is necessary to rule out this gRNA-only mechanism. Lastly, ORF#1 gRNA showed a negligible effect with dPspCas13b-NES (Fig. 1F) but a strong effect with dPspCas13b-NLS (Fig. S1B) (statistical tests are needed here). While ORF-targeting gRNA is not the focus in this study, this unexpected observation should be discussed in the text.

Related to the mechanism of action: no western blots are shown for any of the targets tested. The decrease of target protein level thus are all indirect, based on either decreased luciferase activity (which could be affected by protein misfolding), decreased GFP translation efficiency (which seems to be partially explained by an increase of GFP mRNA, Fig. 3D), or using HiBiT assays. I'm not familiar with HiBiT, but it appears no denaturing step is involved, thus it may not detect aggregated dipeptide proteins. I recommend western blots for the luciferase reporters and GFP reporters, as well as endogenous targets (see comment below).

Generalizability of targeting endogenous mRNAs: Only one endogenous target (CD46) has been tested, and the efficiency is moderate (~50%). To demonstrate the utility of this approach, I strongly recommend testing the approach on a few more endogenous mRNAs of varying abundance. Target abundance is likely a key determinant of the efficiency, as each target mRNA requires stable binding of a dCas13 molecule to inhibit translation. I also suggest to target non-cell surface proteins and use western blot as readout, as cell surface proteins are translated differently from cytoplasmic proteins.

Statistical tests and P values are missing for most bar plots. This includes some key claims, such as Fig. 1D intended to show a lack a change of mRNA abundance.

Minor comments

Fig. 2E: Please double check this plot. In particular, I couldn't find the gRNA that shows 0.8 repression (y-axis) but 0.2 cutting (x-axis) in Fig 2B,D/S2. Please include it in a supplementary figure. I also recommend including the raw data (gRNA sequence and efficiency) in a supplementary table.

Fig. 5B: given the relatively weak effect at the protein level, I'd like to see the lack of change at the RNA level, to confirm the effect is due to translational inhibition.

Fig. 5B: does overexpressing the G4C2 repeat reporter RNA cause cellular toxicity? If so can it be rescued with dCas13b translational inhibition? Given the overall moderate knockdown throughout the study, it would be strengthened if there is some functional impact of the knockdown.

Fig. 5B: If space allows, would be good to discuss two interesting observations here. First, start codon (CUG) gRNA has no effect on translation for any frame. Could this be due to initiation at upstream CUGs? Such as the one found 9nt upstream of the annotated CUG. Second, it appears that the loss of dipeptide repeat in one frame is being compensated by another frame. As it appears that most gRNAs have two DPRs with a reduction while a third leads to overexpression/no change (Figure S5B).

Fig. 6E vs Fig. 6F: looking at the control samples, it appears CD46 is down-regulated by the 4EHP fusion compared to dCas13 alone, suggesting a potential gRNA-independent, global down-regulation of translation caused by 4EHP overexpression, which is expected given 4EHP's known role in inhibiting cap-dependent translation. However, Fig. S6C showed there is no global change of nascent translation. One potential explanation is that the OP-puro assay was normalized by total protein content, which will remove any effect on global translation, given that 4EHP has been expressed for 48 hours prior to OP-puro labeling. Instead of normalizing to total protein content of each sample, the authors should normalize to cell counts. Again 4EHP over-expression is expected to inhibit global translation. It does not affect the authors' conclusion that with a start codon targeting gRNA, dCas13-4EHP fusion results in a stronger inhibitory effect compared to dCas13 alone.

Reviewed by Xuebing Wu, with assistance from Michael Murphy, an Associate Research Scientist in the Wu lab.

Reviewer #3:

Remarks to the Author:

This study develops CRISPR δ , a translation silencing platform that relies on nuclease dead Cas13 enzyme brought to the start codon of mRNAs. This platform blocks ribosome initiation and protein production without mRNA degradation. Importantly, the system has shockingly low offtarget effect on the levels of transcription and translation. The system can be used for canonical cap-mediated, IRES, and RAN driven translation, showing its versatility. Finally, the authors improve the efficiency of the system by fusing the translation inhibitor 4EHP to the Cas13.

This system is novel and will be really useful for the field. The manuscript is well written, and easy to follow. My only concern is that the experiments described are limited to a single cell line and overwhelmingly engineered reporters vs endogenous messages. I strongly believe that the authors need to do more thorough characterization of the system before publication. This additional data will only make the manuscript stronger and more useful for the field.

Needed experiments:

1. Comparison between different cell lines – the authors focus on a single cell line (HEK293). To strengthen their argument, the authors need to explore different cell lines – adherent vs suspension, primary vs transformed, etc.
2. Endogenous mRNAs – there is a single experiment targeting an endogenous mRNA in Fig. 6. The authors need to explore how efficient their system is at blocking translation of mRNAs that differ in 1) abundance, 2) translation efficiency, 3) 5' UTR length.
3. Control experiments using sgRNA alone without Cas13 – the authors never address the possibility that the sgRNA alone will cause a response by either base pairing with the start codon and preventing recognition by the ribosome or by the RNAi pathway. This experiment is key for data interpretation.

General comments by figure:

Fig 1

1. I do not understand the reporter(s) used – do the authors transfect/transduce both Fluc and Rluc on separate mRNAs? Are they on the same mRNA but separated by IRES?
2. Can the authors order 1C so that Cas13 species are next to each other and there is NES/NLS order? The figure is scrambled for no obvious reason and is difficult to read and interpret
3. The authors make a point about NES/NLS, but never actually visualize the Cas13 constructs. Can the authors provide microscopy data to confirm subcellular localization or remove that comparison since the nuclear localization signal does not make a difference for the variant they chose to follow?

Fig. 2

1. Can the authors describe in more detail the sgRNA? Does it have a constant region, what does it look like? Is there a PAM requirement, similar to Cas9 that most of the scientific world is familiar with?

Fig. 3

1. Amazing!

Fig. 4

1. Again, I am unclear on the construct. Are the authors using cap-Fluc-IRES-Rluc? Are these two independent mRNAs? The proper experiment is to have Fluc and Rluc on the same mRNA, as Fluc is under cap-dependent translation, whereas Rluc is under IRES-dependent translation.
2. Can the authors speculate on whether other types of IRESes will be more or less susceptible to CRISPR δ inhibition? Maybe IRESes that require cellular machinery and undergo conformational changes will be able to displace Cas13?

Fig. 5

1. Panel C is impossible to read. I had to print it and use a ruler to figure out what bar corresponds to what condition. The authors should keep the 3 different reading frames as groups, and the conditions in order.

Fig. 6

1. Does recruitment of 4EHP-dCas13 affect mRNA levels?
2. 4EHP is a potent inhibitor that has been hypothesized to recruit mRNAs into an inhibitory environment. This can explain the observation that the 4EHP fusion can inhibit translation even when recruited to the 3'UTR.

Reviewer #1

Apostolopoulos et al. describe an innovative use of deactivated Cas13, termed CRISPRdelta, to inhibit translation in mammalian cells, but not affect mRNA levels. The authors provide very strong data to support their conclusions and should be commended for the overall systematic dissection they performed. The authors first test multiple different Cas13 homologs to identify that dPspCas13-NES is most effective and then proceed to testing and optimizing the gRNA that targets the start codon of luciferase. The rules learned here did generally apply to other mRNAs, but was less effective against the GGGGCC repeat and RAN translation; however, this is perhaps expected with much of the mechanism of GGGGCC RAN translation still unknown and that a single repeat mRNA is translated in multiple reading frames. In nearly every experiment, the authors also include the active PspCas13b as a control to demonstrate that the gRNAs in theory do function and can recognize the target. This was important because it gives validity to the read out of the translational repression assays with dPspCas13b and the many gRNAs tested. The ribosome profiling in GFP expressing stable lines demonstrates that only WT PspCas13, and not dPspCas13, has collateral effects. The manuscript ends on the ability of CRISPRdelta to target endogenous mRNAs, which was extremely important for the authors to demonstrate. In total, the manuscript is focused, well-written, novel, and would be of general interest.

We thank Reviewer #1 for his or her careful reading of the manuscript and for the positive comments.

I have four major critiques for the authors:

1) It is clearly demonstrated that CRISPRdelta is effective against reporter mRNAs during transient co-transfection in HEK293 cells (~5-fold reduction in Luc activity). The experiments testing transcripts originating from a chromosome (e.g, the stable GFP line in Figure 3 or CD46 in Figure 6) appear to be less robust (e.g., ~2-fold reduction in the GFP in the stable line in cells that got the most mCherry-tagged dCas13), especially in the light that RNAi with commercial siRNAs produce nearly 100% knockdown of the target protein.

We thank the reviewer for pointing this out. The efficiency is one of the caveats of the current CRISPR δ approach. Compared to those of RNAi-mediated knockdown, the efficiency and specificity of these methods are now in trade-offs: RNAi, high efficiency but moderate specificity; CRISPR δ , high specificity but moderate efficiency. We highlighted this issue as “Limitations” in the Discussion section.

I cannot tell if the assay in Figure 6 used the entire cell population or just the highly transfected pool like in Figure 3. This certainly may be due to the transfection/expression pattern one often gets from transient plasmid transfection.

We apologize for the unclear explanation of the experiments conducted in Figure 6C-G. As the reviewer suggested, the data in Figure 6C-G were obtained from the cell

subpopulation in which PspCas13b variant expression was ensured. We added more concrete explanations to the text.

My advice is two-fold: i) Unlike plasmid transfections, mRNA transfections with Lipofectamine MessengerMAX, at least in our hands, gives robust, even, and complete transfection. My thought is using this delivery method would provide a more dramatic effect when assayed by Western or FACS; ii) using the more efficacious fused translational repressor as in Figure 6 in addition to the point directly above could provide more robust repression for endogenous targets.

With that being said, this critique would be voided if the authors can provide a Western blot showing >5-fold reduction for CD46 in the total cell population (related #2 below).

We thank the reviewer for the constructive suggestion. Accordingly, we compared dCas13 expression between plasmid DNA and *in vitro*-prepared mRNA (see the figure below and Figure S3B). We observed uniform expression of dCas13 variants among cells by mRNA transfection. On the other hand, DNA transfection generated a variety of dCas13 expression; a subpopulation of cells highly expressed dCas13, allowing translational repression of the target mRNAs. Improving dCas13 expression is an apparent challenge that needs to be addressed in the future. We highlighted this issue as “Limitations” in the Discussion section.

2) Since most users of CRISPRdelta would realistically use Westerns (and not ribosome profiling or FACS) to confirm translational repression and probably not easily be able to use FACS in their desired experiments, it would be highly valuable to demonstrate reduction of a targeted endogenous protein by Western blot. CD46 would suffice of course. We have found that siRNA knockdown of GAPDH is doable after 48 hrs; so that is another option. I believe any endogenous protein would suffice and prove the validity of the technique to most future users.

According to the reviewer’s suggestion, we conducted additional experiments to ensure the reduction of proteins via antibody-based detection. Due to the stochastic nature and low expression of Cas13 variants in cells (Figure S3B), we sorted cells to ensure Cas13 variant expression, as conducted in ribosome profiling (Figure 3B-D and Figure 6C-G).

We performed a Western blot for EGFP (see the figure below and Figures 3E and

S3F), which was expressed from a genome-integrated locus. We observed a clear reduction in the target protein level following the targeting of the start codon by both WT PspCas13 and dPspCas13.

Moreover, we conducted immunohistochemistry (IHC), another antibody-based method, and FACS-mediated detection of ANXA4, a calcium/phospholipid-binding protein, and monitored the translational repression of ANXA4 with CRISPR δ (see the figure below and Figure S6G-I).

As mentioned above and below, improving Cas13 variant expression is the next venue for the wider use of CRISPR δ . We highlighted this issue as “Limitations” in the Discussion section.

3) I apologize if I missed it, it appears all experiments have been performed in HEK293 cells. It is important to provide evidence that CRISPRdelta would be nearly as functional in another cell type. To be clear, this is NOT a request to repeat everything in a different cell type. I recommend a robust and well controlled experiment shown in Figure 1E or Figure 6B.

According to the reviewer's suggestion, we repeated key experiments with HeLaS3 suspension cells and mouse embryo fibroblast (MEF) primary culture (see the figure below and Figure S2I-J). We observed repression of reporter genes with dPspCas13b.

4) It would be very strong to demonstrate mRNA levels are not changing in Figure 6 when using the fused translational repression version, for both the luciferase and CD46. The OPP measurement was great and does provide evidence to the readers that global translation is not affected.

According to the reviewer's comments, we conducted RT-qPCR for target mRNAs (exogenous Rluc or endogenous CD46) with dPsp-Cas13b-4EHP recruitment (see the figures below and Figure S6C and S6F). We did not observe a significant change in the expression of these mRNAs under these conditions.

Minor critiques:

5) The authors mention at multiple locations about the tight binding of Cas13 to mRNA, but they make the readers look into previous literature for what this really is. Can the authors include reported affinity measurements in the text?

We thank the reviewer for the constructive suggestion. We included a summary of K_d values (5-40 nM) from several reports (Abudayyeh *et al. Science* 2016; East-Seletsky *et al. Nature* 2016; Smargon *et al. Mol Cell* 2017; Tambe *et al. Cell Rep* 2018; Yang *et al. Nat Chem Biol* 2023) in the manuscript.

6) In Figure 1B, are all Cas13 homologs equally expressed? This could be an explanation to why only some of the WT homologs are functional. This is not a major point since the main focus of the paper is to demonstrate that at least a few can be used to repress translation.

According to the reviewer's suggestion, we conducted a Western blot for the Cas13 protein variants (see the figure below and Figure S1C). We observed differences in the expression levels of Cas13 variants. The localization tags also affected the protein abundance. On the other hand, the catalytic activity of Cas13 did not strongly impact the protein level. Thus, the results in Figure 1B included the effects of not only the efficacy of mRNA cleavage by WT variants and translation repression by catalytically dead variants but also the efficacy of protein expression. We added this note to the manuscript.

Although NES-tagged RfxCas13d was the most inefficient at gene silencing (Figure 1C left), the expression of this protein was not the lowest. In contrast, NES-tagged dPspCas13b was most inefficiently expressed but exhibited the greatest translational repression (Figure 1C right). Thus, protein expression explained the knockdown efficiency only partially.

7) It may clearer to keep the “translational repression” nomenclature instead of “gene knockdown” when referring to the CRISPRdelta since nearly the entire field always uses knockdown when describing RNAi.

According to the reviewer’s suggestion, we avoided the wording “gene knockdown” to refer to CRISPR δ throughout the manuscript.

8) Rationale is not entirely laid out for testing the NES and NLS versions in Figure 1C.

Since earlier reports (Abudayyeh *et al.* Nature 2017; Cox *et al.* Science 2017; Tong *et al.* Nat. Biotechnol. 2023; Wilson *et al.* Nat. Biotechnol. 2020; Wessels *et al.* Nat. Biotechnol. 2020) have shown the differential potential of Cas13-mediated tools due to its localization in the cytoplasm and nucleus, we were encouraged to test NES and NLS variants. NLS tagging of Cas13 variants may be advantageous for complex formation with transcribed gRNA before degradation, while NES tagging should be more reasonable for inhibiting translation, which occurs in the cytoplasm. Indeed, RfxCas13d required the NLS to be more functional (Figure 1C). We added a detailed explanation of our motivation for the investigation of NES/NLS tags.

Related to this, NLS-tagged dPspCas13b may partially sequester the target mRNA into the nucleus, inhibiting accessibility to the translation machinery. We highlighted this point as “Limitations” in the Discussion section.

Nor is it demonstrated, even on dPspCas13b, they are functional on control transfections with no gRNA.

According to the reviewer’s suggestion, we conducted experiments with only Cas13 proteins (see the figure below and Figure S1E). As expected, no repression was induced by just the Cas13 protein.

9) In the methods for CD46 expression via FACS, it is not specified what cell type was used. I assume HEK293 cells again.

According to the reviewer's suggestion, we specified the cell type in the Method section (as the reviewer guessed, we used HEK293 cells for this experiment).

Reviewer #2

In the manuscript titled “CRISPR δ : dCas13-mediated translational repression for accurate gene silencing in mammalian cells”, the authors developed a dCas13-based method to inhibit the protein production from a specific target mRNA without affecting mRNA abundance. This approach also avoids the collateral activity associated with wild type Cas13-based RNA knockdown. This study not only provides a new tool for loss-of-function studies of protein-coding genes, but also opened the door for studying protein-independent function of coding mRNAs. The method represents an important new application of RNA-targeting CRISPR systems in mammalian cells. The research is novel considering the differences between bacterial and mammalian expression systems.

Overall, the experiments were carefully designed and carried out by an experienced group. The authors systematically optimized several parameters of the system, including Cas13 effectors, subcellular localization of effectors (NLS vs NES), gRNA length, target position, etc. The system has been tested with luciferase reporters (Cap-dependent and IRES-dependent), a GFP reporter, a disease-relevant reporter (G4C2 repeat), and an endogenous target (CD46). The specificity of the approach is supported by compelling RNA-seq and ribosome-profiling data.

I strongly support the publication of this study, if the authors can address a few concerns detailed below.

We thank Reviewer #2 for his or her careful reading of the manuscript and for the positive comments.

Major comments:

Mechanism of action: while steric hindrance of translation initiation at the start codon remains the most likely explanation for the observed decrease in protein activity/abundance, several observations are not consistent with this model and will need clarifications or control experiments to rule out alternative mechanisms. For example, there is a negligible difference between NLS- and NES-containing dCas13b at the start codon (Fig. 1C). One would expect an NLS-containing dCas13b to be unable to influence translation. One possibility is that NLS-dCas13b is loaded in the nucleus and then exported together with the mRNA to the cytoplasm, which then inhibits translation initiation. Or, NLS-dCas13b may inhibit translation via a different mechanism, e.g., sequestering target mRNAs in the nucleus and prevents translation. A discussion on these possibilities is recommended.

We agree with the reviewer’s suggestion regarding the nuclear sequestration of target mRNAs by NLS-tagged dCas13 variants. However, since NES-tagged dPspCas13b showed the strongest translational repression in the first screening (Figure 1C), we mainly focused on the variant in this study. Although the potential for nuclear sequestration of mRNAs is reasonable, this point is beyond the scope of this study. Instead, we highlighted this issue as “Limitations” in the Discussion section.

Another alternative model that needs to be excluded is that, dCas13b may not be required for target inhibition, i.e., the gRNA itself is sufficient to induce inhibition via steric hindrance, RNA editing, or RNAi-like effect. The inefficiency of other Cas13 proteins does not rule out this possibility, as the scaffold regions are different for each subtype of Cas13 system. A control experiment in which dPspCas13b is omitted is necessary to rule out this gRNA-only mechanism.

According to the reviewer's suggestion, we conducted experiments with only gRNA (see the figure below and Figure S1D). As expected, no repression was induced by only gRNA.

Lastly, ORF#1 gRNA showed a negligible effect with dPspCas13b-NES (Fig. 1F) but a strong effect with dPspCas13b-NLS (Fig. S1B) (statistical tests are needed here). While ORF-targeting gRNA is not the focus in this study, this unexpected observation should be discussed in the text.

We thank the reviewer for pointing this out. A possible scenario could be that NLS-tagged dPspCas13b may partially sequester the target mRNA into the nucleus, as the reviewer suggested, inhibiting accessibility to the translation machinery. We conducted a microscopic analysis of the dPspCas13 variants with an NES or NLS (see the figure below and Figure S1I). The NLS enriched dPspCas13 in the nucleus, allowing a fraction to leak into the cytoplasm. Thus, NLS-tagged dPspCas13 may have dual functions in translational repression: steric hindrance for scanning by a fraction leaked in the cytoplasm and physical sequestration by a fraction enriched in the nucleus. We highlighted this point as "Limitations" in the Discussion section. We also added statistics to Figure S1F (previously Figure S1B).

Related to the mechanism of action: no western blots are shown for any of the targets tested. The decrease of target protein level thus are all indirect, based on either decreased luciferase activity (which could be affected by protein misfolding), decreased GFP translation efficiency (which seems to be partially explained by an increase of GFP mRNA, Fig. 3D), or using HiBiT assays. I'm not familiar with HiBiT, but it appears no denaturing step is involved, thus it may not detect aggregated dipeptide proteins. I recommend western blots for the luciferase reporters and GFP reporters, as well as endogenous targets (see comment below).

Generalizability of targeting endogenous mRNAs: Only one endogenous target (CD46) has been tested, and the efficiency is moderate (~50%). To demonstrate the utility of this approach, I strongly recommend testing the approach on a few more endogenous mRNAs of varying abundance. Target abundance is likely a key determinant of the efficiency, as each target mRNA requires stable binding of a dCas13 molecule to inhibit translation. I also suggest to target non-cell surface proteins and use western blot as readout, as cell surface proteins are translated differently from cytoplasmic proteins.

According to the reviewer's suggestion, we conducted additional experiments to ensure the reduction of proteins via antibody-based detection. Due to the stochastic nature and low expression of Cas13 variants in cells (Figure S3B), we sorted cells to ensure Cas13 variant expression, as conducted in ribosome profiling (Figure 3B-D and Figure 6C-G).

We performed a Western blot for EGFP (see the figure below and Figures 3E and S3F), which was expressed from a genome-integrated locus. We observed a clear reduction in the target protein level following the targeting of the start codon by both WT PspCas13 and dPspCas13.

Moreover, we conducted immunohistochemistry (IHC), another antibody-based method, and FACS-mediated detection of ANXA4, a calcium/phospholipid-binding protein, and monitored the translational repression of ANXA4 with CRISPR δ (see the figure below and Figure S6G-I).

Overall, the results of the present study confirmed the potency of CRISPR δ to the genes at intermediate to high expression levels (see the figure below and Figure S6J).

This efficiency is one of the caveats of the current CRISPR δ approach. Compared to those of RNAi-mediated knockdown, the efficiency and specificity of these methods are now in trade-offs: RNAi, high efficiency but moderate specificity; CRISPR δ , high specificity but moderate efficiency. We highlighted this issue as “Limitations” in the Discussion section.

Statistical tests and P values are missing for most bar plots. This includes some key claims, such as Fig. 1D intended to show a lack a change of mRNA abundance.

According to the reviewer's suggestion, we added p values to the data, especially where we highlighted the difference in outputs (e.g., mRNA degradation vs. its absence, active gRNA vs. inactive gRNA, etc.).

Minor comments:

Fig. 2E: Please double check this plot. In particular, I couldn't find the gRNA that shows 0.8 repression (y-axis) but 0.2 cutting (x-axis) in Fig 2B,D/S2. Please include it in a supplementary figure. I also recommend including the raw data (gRNA sequence and efficiency) in a supplementary table.

The data point corresponding to "0.8 repression (y-axis) but 0.2 cutting (x-axis)" should be gRNA 5' UTR #5 in Figure 1H. We clarified the source data/figure panel for Figure 2E in the legend. Moreover, according to the reviewer's suggestion, we added Table S2 (and Source Data) for the raw data and gRNA sequences.

Fig. 5B: given the relatively weak effect at the protein level, I'd like to see the lack of change at the RNA level, to confirm the effect is due to translational inhibition.

We truly thank the reviewer for pointing this out and providing an opportunity to improve our manuscript. According to the reviewer's suggestion, we conducted RT-qPCR for the poly(GA) reporter with the PspCas13b variant and gRNA of 5' UTR#1 (see the figure below and Figure S5E). Unexpectedly, in addition to WT PspCas13b, dPspCas13b also induced a reduction of target mRNAs. Within the set of conditions examined (Figures 1D, 3D, S6C, and S6F), this phenomenon was the sole exception. Since DPRs synthesized from *C9orf72* GGGGCC repeats suppress RNA exosome activity and stabilize transcripts (Kawabe *et al.*, *EMBO J* 2020), dCas13-mediated translational repression may lead to destabilization of mRNAs with GGGGCC repeats. We added a detailed explanation to the text and highlighted this exception.

Fig. 5B: does overexpressing the G4C2 repeat reporter RNA cause cellular toxicity? If so can it be rescued with dCas13b translational inhibition? Given the overall moderate knockdown throughout the study, it would be strengthened if there is some functional

impact of the knockdown.

According to the reviewer's suggestion, we tested whether the GGGGCC repeat reporter affects cell viability (see the figure below and Figure S5A). Our reporter did not induce severe cell toxicity, probably due to its mild expression. Thus, unfortunately, we could not perform the experiments suggested by the reviewer. We are actively working on the application of CRISPR δ to motor neurons differentiated from patient-derived iPSCs to investigate its neuroprotective potential. However, this topic was beyond the scope of this study.

Fig. 5B: If space allows, would be good to discuss two interesting observations here. First, start codon (CUG) gRNA has no effect on translation for any frame. Could this be due to initiation at upstream CUGs? Such as the one found 9nt upstream of the annotated CUG. Second, it appears that the loss of dipeptide repeat in one frame is being compensated by another frame. As it appears that most gRNAs have two DPRs with a reduction while a third leads to overexpression/no change (Figure S5B).

We appreciate the reviewer's constructive suggestion. Consistent with the reviewer's thoughts, earlier works reported a complicated outcome when the CUG codon was mutated (Green *et al.* Nat. Commun. 2017; Sonobe *et al.* Neurobiol. Dis. 2018; Tabet *et al.* Nat. Commun. 2018; Boivin *et al.* EMBO J. 2020; Lampasona *et al.* Neurobiol. Aging 2021; van 't Spijker *et al.* RNA 2022). Even with extensive efforts, the RAN translation mechanism for *C9orf72* GGGGCC repeats has been debated.

One clear consensus of these studies was that poly(GA), the most abundant dipeptide repeat protein (DPR), is driven by CUG translation initiation sites (Green *et al.* Nat. Commun. 2017; Sonobe *et al.* Neurobiol. Dis. 2018; Tabet *et al.* Nat. Commun. 2018; Boivin *et al.* EMBO J. 2020; van 't Spijker *et al.* RNA 2022; Latallo *et al.* Nat Commun 2023).

In contrast, multiple different scenarios for the translation initiation of poly(GR), which is -1 frame to poly(GA), were proposed. Several studies suggested that the same CUG translation initiation site was used for poly(GA) but followed by -1 frameshift to generate poly(GR) (Green *et al.* Nat. Commun. 2017; Tabet *et al.* Nat. Commun. 2018; van 't Spijker *et al.* RNA 2022; Latallo *et al.* Nat Commun 2023). However, other studies

have suggested that poly(GR) may be independent of CUG start codons (Lampasona *et al.* Neurobiol. Aging 2021) and rather require an in-frame AGG subcognate initiation codon (Boivin *et al.* EMBO J. 2020), which is located downstream of the CUG.

The translation initiation of poly(GP) is even more complicated. Tabet *et al.* suggested CUG-initiated translation and subsequent +1 frameshift (Tabet *et al.* Nat. Commun. 2018; Latallo *et al.* Nat Commun 2023). However, this effect was condition-dependent since some types of cells still produced poly(GP) from the reporter even without the CUG codon, suggesting a shift of translation initiation sites (Tabet *et al.* Nat. Commun. 2018). Related to this, Green *et al.* reported that poly(GP) did not use the CUG or AGG codons for initiation (Green *et al.* Nat. Commun. 2017). Poly(GP) translation could be influenced by the translation of other DPRs since deletion of the CUG codon even enhances poly(GP) production (Green *et al.* Nat. Commun. 2017; van 't Spijker *et al.* RNA 2022).

As the reviewer suggested, we also first considered harnessing our system to probe the translation initiation sites of *C9orf72* GGGGCC repeats. However, given the cell type-dependent and condition-dependent fluctuations in the results [especially for poly(GR) and poly(GP)] and the potential shift of initiation sites reflected by other DPR production statuses, we avoided using our data to interpret translation initiation sites of DPRs and to discuss the complexity of translation initiation sites. However, we are happy to add a detailed explanation to the text when the reviewer and editor think it necessary.

Fig. 6E vs Fig. 6F: looking at the control samples, it appears CD46 is down-regulated by the 4EHP fusion compared to dCas13 alone, suggesting a potential gRNA-independent, global down-regulation of translation caused by 4EHP overexpression, which is expected given 4EHP's known role in inhibiting cap-dependent translation. However, Fig. S6C showed there is no global change of nascent translation. One potential explanation is that the OP-puro assay was normalized by total protein content, which will remove any effect on global translation, given that 4EHP has been expressed for 48 hours prior to OP-puro labeling. Instead of normalizing to total protein content of each sample, the authors should normalize to cell counts. Again 4EHP over-expression is expected to inhibit global translation. It does not affect the authors' conclusion that with a start codon targeting gRNA, dCas13-4EHP fusion results in a stronger inhibitory effect compared to dCas13 alone.

We appreciate the reviewer's constructive suggestion. To address the reviewer's concern about the effect of 4EHP-fused Cas13 variants on cell growth, we tested the viability of the cells. As shown in the figure below and in Figure S6E, we did not observe any impact on cell growth under our conditions, possibly due to the low level of expression. The artificial recruitment of 4EHP allows the protein access to the cap structure of target mRNAs even at such a low concentration of 4EHP.

Reviewed by Xuebing Wu, with assistance from Michael Murphy, an Associate Research Scientist in the Wu lab.

Again, we thank Professor Wu and Dr. Murphy for their careful reading of our manuscript and many constructive suggestions.

Reviewer #3

This study develops CRISPR δ , a translation silencing platform that relies on nuclease dead Cas13 enzyme brought to the start codon of mRNAs. This platform blocks ribosome initiation and protein production without mRNA degradation. Importantly, the system has shockingly low offtarget effect on the levels of transcription and translation. The system can be used for canonical cap-mediated, IRES, and RAN driven translation, showing its versatility. Finally, the authors improve the efficiency of the system by fusing the translation inhibitor 4EHP to the Cas13.

This system is novel and will be really useful for the field. The manuscript is well written, and easy to follow. My only concern is that the experiments described are limited to a single cell line and overwhelmingly engineered reporters vs endogenous messages. I strongly believe that the authors need to do more thorough characterization of the system before publication. This additional data will only make the manuscript stronger and more useful for the field.

We thank Reviewer #3 for his or her careful reading of the manuscript and for the positive comments.

Needed experiments:

1. Comparison between different cell lines – the authors focus on a single cell line (HEK293). To strengthen their argument, the authors need to explore different cell lines – adherent vs suspension, primary vs transformed, etc.

According to the reviewer's suggestion, we repeated key experiments with HeLaS3 suspension cells and mouse embryonic fibroblast (MEF) primary culture (see figure below and Figure S2I and S2J). We observed repression of reporter genes with dPspCas13b.

2. Endogenous mRNAs – there is a single experiment targeting an endogenous mRNA in Fig. 6. The authors need to explore how efficient their system is at blocking translation of mRNAs that differ in 1) abundance, 2) translation efficiency, 3) 5' UTR length.

Currently, this study covered 3 examples of endogenous genes (including the one integrated into the genome): EGFP, ANXA4, and CD46. Based on the ribosome profiling

data, which correspond to overall protein synthesis, our data confirmed the ability of CRISPR δ to the genes at intermediate to high expression levels (see the figure below and Figure S6J). Given that dCas13 is required to be stoichiometrically abundant to mRNA and to ribosome flux on the mRNA, these data ensure the wide application of CRISPR δ for gene silencing.

These mRNAs do not have extreme differences in the length of the 5' UTR or in translation efficiency (TE) (see the figures below). Large-scale investigations of gRNAs or target identities will further clarify the potential of CRISPR δ for mRNAs with long/short 5' UTRs and high/low TEs. However, this topic is beyond the scope of this study.

3. Control experiments using sgRNA alone without Cas13 – the authors never address the possibility that the sgRNA alone will cause a response by either base pairing with the start codon and preventing recognition by the ribosome or by the RNAi pathway. This experiment is key for data interpretation.

According to the reviewer's suggestion, we conducted gRNA-only experiments (see the

figure below and Figure S1D). As expected, no repression was induced by only gRNA.

General comments by figure:

Fig 1

1. I do not understand the reporter(s) used – do the authors transfect/transduce both Fluc and Rluc on separate mRNAs? Are they on the same mRNA but separated by IRES?

We apologize for the unclear explanation. We transfected a DNA plasmid encoding both RLuc and Fluc (psiCHECK2, Promega). They are transcribed into two different mRNAs whose translation is driven by a canonical cap-dependent mechanism. We clarified this point in the manuscript and prepared Figure S1A (and see also the figure below) for the schematic representation.

2. Can the authors order 1C so that Cas13 species are next to each other and there is NES/NLS order? The figure is scrambled for no obvious reason and is difficult to read and interpret

In the original panel, the data were ordered following the repression efficiency found for dCas13. According to the reviewer's suggestion, we reformatted this panel.

3. The authors make a point about NES/NLS, but never actually visualize the Cas13

constructs. Can the authors provide microscopy data to confirm subcellular localization or remove that comparison since the nuclear localization signal does not make a difference for the variant they chose to follow?

According to the reviewer’s suggestion, we conducted a microscopic analysis of the dPspCas13 variants with an NES or NLS (see the figure below and Figure S1I). The NLS enriched dPspCas13 in the nucleus, allowing a fraction to leak into the cytoplasm. Thus, NLS-tagged dPspCas13 may have dual functions in translational repression: steric hindrance for scanning by a fraction leaked in the cytoplasm and physical sequestration by a fraction enriched in the nucleus. We highlighted this point as “Limitations” in the Discussion section.

Fig. 2

1. Can the authors describe in more detail the sgRNA? Does it have a constant region, what does it look like? Is there a PAM requirement, similar to Cas9 that most of the scientific world is familiar with?

According to the reviewer’s suggestion, we added detailed sequence/structural information (especially for the DR region) to Figure S1B (see also the figure below).

Cas13 gRNAs do not require a PAM and have minimal restriction of the protospacer flanking sequence (PFS) in human cells (Abudayyeh *et al.*, *Science* 2016; Smargon *et al.*, *Mol Cell* 2017; Abudayyeh *et al.*, *Nature* 2017; Cox *et al.*, *Science* 2017; Konermann *et al.*, *Cell* 2018; Meeske *et al.*, *Mol Cell* 2018). We clarified this point in the manuscript.

Fig. 3

1. Amazing!

We thank the reviewer for the positive comments.

Fig. 4

1. Again, I am unclear on the construct. Are the authors using cap-Fluc-IRES-Rluc? Are these two independent mRNAs? The proper experiment is to have Fluc and Rluc on the same mRNA, as Fluc is under cap-dependent translation, whereas Rluc is under IRES-dependent translation.

According to the reviewer's suggestion, we added Figure S4A and S4C (see also the figure below) for the schematic representation of DNA construction for this assay.

Given the DNA transfection, HCV-IRES-Rluc mRNA synthesized in cells should have an m⁷G cap and poly(A) tail. Nonetheless, HCV-driven translation dominated overall translation rather than cap-dependent translation, as evidenced by resistance to hippuristanol, an eIF4A inhibitor. Fluc mRNA, an internal control, was synthesized as an independent transcript.

On the other hand, we found that a similar DNA construction for the CrPV IGR IRES (related to the point below) was not suitable since the reporter mRNA generated from the DNA plasmid was capped by endogenous enzymes and was mainly driven by a cap-dependent (at least hippuristanol sensitive) mechanism (data not shown). For this reason, we prepared A-capped mRNA *in vitro* and transfected it directly into cells. Although we did not use Fluc for normalization in this experiment, we observed reproducible results (Figure 4C-D).

2. Can the authors speculate on whether other types of IRESes will be more or less susceptible to CRISPR δ inhibition? Maybe IRESes that require cellular machinery and undergo conformational changes will be able to displace Cas13?

According to the reviewer's comments, we performed an additional assay with the cricket

paralysis virus (CrPV) intergenic region (IGR) IRES, which does not require any translation initiation factors and directly recruits the 40S ribosome (Fernández *et al.*, *Cell* 2014; Murray *et al.*, *eLife* 2016). Moreover, this IRES starts translation at the Ala GCU codon. Even in this extreme case, CRISPR δ can repress translation (see the figure below and Figure 4C-D), indicating the wide application of this methodology for a variety of translation mechanisms. We agree with the reviewer that the secondary structure of IRES and the interacting factors may ultimately affect the efficiency of CRISPR δ , although the prediction of the efficiency is difficult at this moment. We added the discussion regarding this point in the manuscript.

Fig. 5

1. Panel C is impossible to read. I had to print it and use a ruler to figure out what bar corresponds to what condition. The authors should keep the 3 different reading frames as groups, and the conditions in order.

According to the reviewer's suggestion, we added reformatted panels to Figure S5B-D. As a main figure panel, we placed the rank plot of the same data to highlight differential efficiencies of the gRNAs.

Fig. 6

1. Does recruitment of 4EHP-dCas13 affect mRNA levels?

According to the reviewer's comments, we conducted RT-qPCR for target mRNAs (exogenous Rluc or endogenous CD46) with dPsp-Cas13b-4EHP recruitment (see the figures below and Figure S6C and S6F). We did not observe a significant change in the expression of these mRNAs under these conditions.

2. 4EHP is a potent inhibitor that has been hypothesized to recruit mRNAs into an inhibitory environment. This can explain the observation that the 4EHP fusion can inhibit translation even when recruited to the 3'UTR.

We thank the reviewer for carefully reading our manuscript. We agree with this discussion. As we have cited, artificial tethering of 4EHP to the 3' UTR has been shown to induce translational repression (Peter *et al. Genes Dev* 2017; Hickey *et al. Mol Cell* 2020). We added more explanation for the interpretation, highlighting the similarity to the reported results.

Reviewers' Comments:

Reviewer #1:

Remarks to the Author:

After reading the authors' response to the reviewers and the revised manuscript, I feel the authors have honestly and appropriately addressed the reviewers' critiques to the best of their ability without spending years to develop CRISPR δ v2. While the CRISPR δ does have some significant limitations that would most likely prevent most labs from using it to knockdown endogenous genes/mRNAs, I fully appreciate that this tool has the potential to be further refined in subsequent manuscripts and studies. This body of work is important for the next step of developing more efficacious versions that could be used without needing to FACS sort cells to analyze gene expression. The new Limitations section is also a critical addition by the authors. Importantly, and as I noted in my original review, the work is rigorous.

I was reviewer 1. To clarify, the second part of my critique #8 was referring to showing the NES and NLS signals are functional. I apologize that this was not clear in my original comments. Reviewers 2 & 3 also commented on this and the authors now provide some evidence that the localization signals work as expected.

Reviewer #2:

Remarks to the Author:

All of my previous concerns have been addressed.

Reviewer #3:

Remarks to the Author:

The authors have addressed all of my original concerns and provided sufficient experiments and explanations to further strengthen the manuscript.

Reviewer #1:

After reading the authors' response to the reviewers and the revised manuscript, I feel the authors have honestly and appropriately addressed the reviewers' critiques to the best of their ability without spending years to develop CRISPR δ v2. While the CRISPR δ does have some significant limitations that would most likely prevent most labs from using it to knockdown endogenous genes/mRNAs, I fully appreciate that this tool has the potential to be further refined in subsequent manuscripts and studies. This body of work is important for the next step of developing more efficacious versions that could be used without needing to FACS sort cells to analyze gene expression. The new Limitations section is also a critical addition by the authors. Importantly, and as I noted in my original review, the work is rigorous.

I was reviewer 1. To clarify, the second part of my critique #8 was referring to showing the NES and NLS signals are functional. I apologize that this was not clear in my original comments. Reviewers 2 & 3 also commented on this and the authors now provide some evidence that the localization signals work as expected.

We thank Reviewer #1 for his or her careful reading of the manuscript and for the positive comments.

Reviewer #2 (Remarks to the Author):

All of my previous concerns have been addressed.

We thank Reviewer #2 for his or her careful reading of the manuscript and for the positive comments.

Reviewer #3 (Remarks to the Author):

The authors have addressed all of my original concerns and provided sufficient experiments and explanations to further strengthen the manuscript.

We thank Reviewer #3 for his or her careful reading of the manuscript and for the positive comments.